# Preconditioned Neural Posterior Estimation for Likelihood-free Inference

**Xiaoyu Wang**                                                    *x311.wang@hdr.qut.edu.au*
*School of Mathematical Sciences*
*Centre for Data Science*
*Queensland University of Technology*

**Ryan P. Kelly**                                                  *r21.kelly@hdr.qut.edu.au*
*School of Mathematical Sciences*
*Centre for Data Science*
*Queensland University of Technology*

**David J. Warne**                                                 *david.warne@qut.edu.au*
*School of Mathematical Sciences*
*Centre for Data Science*
*Queensland University of Technology*

**Christopher Drovandi**                                           *c.drovandi@qut.edu.au*
*School of Mathematical Sciences*
*Centre for Data Science*
*Queensland University of Technology*

**Reviewed on OpenReview:** *https://openreview.net/forum?id=vgIBAOkIhY*

## Abstract

Simulation-based inference (SBI) methods enable the estimation of posterior distributions when the likelihood function is intractable, but where model simulation is feasible. Popular neural approaches to SBI are neural posterior estimation (NPE) and its sequential version (SNPE). These methods can outperform statistical SBI approaches such as approximate Bayesian computation (ABC), particularly for relatively small numbers of model simulations. However, we show in this paper that the NPE methods are not guaranteed to be highly accurate, even on problems with low dimension. In such settings the posterior cannot be accurately trained over the prior predictive space, and even the sequential extension remains sub-optimal. To overcome this, we propose preconditioned NPE (PNPE) and its sequential version (PSNPE), which uses a short run of ABC to effectively eliminate regions of parameter space that produce large discrepancies between simulations and data and allow the posterior emulator to be more accurately trained. We present comprehensive empirical evidence that this melding of neural and statistical SBI methods improves performance over a range of examples including a motivating example involving a complex agent-based model applied to real tumour growth data.

## 1 Introduction

Computational models, frequently termed as simulators, are typically governed by stochastic processes. When provided with a set of parameter values, these simulators output synthetic data that inherently capture the stochastic nature of the simulated phenomena. However, a substantial challenge arises in performing posterior inference for the parameters of these simulators, as the corresponding likelihood function is often intractable. An example is agent-based modelling of tumour growth (e.g. Jenner et al. (2020); Aylett-Bullock et al. (2021); Warne et al. (2022)), where cell proliferation, movement and invasion are governed by

probabilistic rules, which depend on key biological parameters. Consequently, standard statistical inference methods that rely on a closed-form expression for the likelihood function are inapplicable in this scenario.

To address this issue, simulation-based inference (SBI) methods have been developed, which approximate the posterior based only on simulations from the model. The most thoroughly examined SBI method in the statistical literature is approximate Bayesian computation (ABC) (Sisson et al., 2018). Advancements in deep neural networks have led to the emergence of neural SBI methods (Cranmer et al., 2020). Their widespread application spans various fields, including biology (Wehenkel et al., 2023), neuroscience (Fengler et al., 2021; West et al., 2021), and astronomy (Mishra-Sharma, 2022; Dax et al., 2021).

Statistical SBI methods, such as ABC methods, are well-developed and boast strong theoretical guarantees of convergence to the true posterior (Beaumont et al., 2009; Blum, 2010; Biau et al., 2015; Lintusaari et al., 2017; Sisson et al., 2018; Beaumont, 2019). ABC approaches compare observed to simulated data using a discrepancy function and prefer parameter values that generate discrepancies below a pre-defined threshold, $\epsilon$. However, often a small value of $\epsilon$ is required to obtain accurate posterior approximations, which can significantly increase the number of model simulations required, and hence the computational cost (Csilléry et al., 2010; Turner & Van Zandt, 2012). Advanced ABC samplers such as adaptive sequential Monte Carlo ABC (SMC ABC, e.g. Sisson et al. (2007); Drovandi & Pettitt (2011)) have been developed to mitigate this issue. However, the efficiency of ABC methods may decrease as $\epsilon$ becomes smaller, requiring a significantly larger number of simulated datasets. Enhancing the efficiency of ABC algorithms remains an active area of research (Lintusaari et al., 2017; Sisson et al., 2018).

With the rapid advances in machine learning methods, more efficient approaches based on neural networks have been developed (Papamakarios et al., 2017; 2021). A popular approach is neural posterior estimation (NPE) and its sequential version, the sequential NPE (SNPE) (Papamakarios & Murray, 2016; Lueckmann et al., 2017; Greenberg et al., 2019). These methods use a set of training pairs of parameter values and simulated datasets to fit a neural conditional density estimator (NCDE), such as a conditional normalizing flow (Rezende & Mohamed, 2015; Winkler et al., 2019; Durkan et al., 2019; Dolatabadi et al., 2020), to approximate the posterior. NPE uses parameter values drawn from the prior, whilst SNPE uses parameter values drawn from previous NPE approximations for a given number of rounds. The idea of the sequential approach is that a more accurate emulator of the posterior can be achieved when more (parameter value, simulated data) training pairs are generated in higher density regions of the posterior. Lueckmann et al. (2021) have shown that NPE can outperform ABC, particularly for a relatively small number of model simulations.

However, in real-world problems, there may be little known about the parameters *a priori*, so that a vague prior may be employed. For example, a uniform distribution with a wide constraint range may be used as a prior distribution. For some problems in this setting we find that it is difficult to construct an accurate NCDE across a wide parameter space, which leads to NPE producing an inaccurate posterior approximation. We find that even SNPE may not be able to recover from such an initially deficient approximation, even with a relatively large number of rounds, and hence model simulations. One way to mitigate this kind of unstable NCDE training is to clip the extreme simulated datasets (Shih et al., 2023; de Santi et al., 2023). However, this approach is ad-hoc and it is not clear how much clipping is required for a given problem and may require extensive experimentation, and each level of clipping requires refitting of the NCDE.

This paper contains three key contributions. Firstly, we explore several examples where NPE methods fail to produce highly accurate posterior approximations, even in relatively low dimensional problems. Our second contribution is the development of preconditioned NPE (PNPE), and its sequential extension (PSNPE), which combines the strengths of statistical and ML approaches to SBI. The preconditioning step involves applying an ABC algorithm for efficiently discarding parts of the parameter space that lead to large discrepancies, which then subsequently permits NPE methods to perform well. In a sense, our preconditioning step acts as a principled clipping method. Our third contribution shows via an extensive empirical study that our preconditioned NPE approaches outperform NPE approaches when the latter performs sub-optimally, and is competitive when it performs well. Our motivating example involves a complex agent-based model applied to real tumour growth data.

## 2    Simulation-based Inference

Consider a simulator that takes parameters $\theta \in \mathbb{R}^d$ where $d$ is the number of parameters and generates a simulated dataset $x \in \mathbb{R}^D$ where $D$ is the dimension of the data, but its density $p(x|\theta)$ is intractable. The objective of SBI is to accurately estimate the posterior density of $\theta$ conditional on the observed dataset $x_o \in \mathbb{R}^D$ based only on simulating data from the model and not requiring evaluation of the intractable likelihood, $p(x_o|\theta)$. Two popular SBI methods are ABC and NPE, which are summarized below.

### 2.1    Approximate Bayesian computation

Statistical SBI (Sisson et al., 2018), such as the ABC rejection algorithm, is based on Monte Carlo rejection sampling. That is, it keeps only the parameter values simulated from the prior that generate simulated data $x$ such that $\rho(x, x_o) < \epsilon$, where $\rho(x, x_o)$ is a user-defined discrepancy function between the simulated and observed data, and $\epsilon$ is a user-defined threshold often referred to as the ABC tolerance.

SMC ABC algorithms (e.g. Sisson et al. (2007); Drovandi & Pettitt (2011)) aim to be more efficient by sampling a sequence of ABC posteriors with decreasing $\epsilon$'s, updating the importance distribution at each iteration. More specifically, SMC ABC algorithms define a sequence of non-increasing ABC thresholds $\epsilon_1 \geq \epsilon_2 \geq \cdots \geq \epsilon_T$, such that

$$p_{\epsilon_t}(\theta|x_o) \propto p(\theta) \int_{\mathbb{R}^D} \mathbb{I}\big(\rho(x_o, x) < \epsilon_t\big) p(x|\theta)dx, \qquad \text{for } t = 1, \ldots, T. \tag{1}$$

Here, $\epsilon_T = \epsilon$ represents the target ABC posterior.

In many real-world applications, $x, x_o \in \mathbb{R}^D$ are considered high-dimensional data, necessitating a mapping to a lower-dimensional space for computational efficiency. This is typically done using summary statistics $S(\cdot)$. If summary statistics are required, we use $S(x)$ and $S(x_o)$ instead of the full datasets $x$ and $x_o$. The choice of appropriate summary statistics is a subject of ongoing research and is discussed extensively in the literature (see Sisson et al. (2018)).

However, even sophisticated ABC algorithms can require a significant number of model simulations to achieve a suitably small value of $\epsilon$ (Biau et al., 2015; Csilléry et al., 2010; Beaumont et al., 2009; Blum, 2010).

### 2.2    Neural posterior estimation

NPE uses $N$ training pairs of simulator parameter values and simulated datasets, $\{\theta_i, x_i\}_{i=1}^N$, to estimate the posterior distribution $p(\theta|x)$ (Papamakarios & Murray, 2016; Lueckmann et al., 2017; Greenberg et al., 2019). Once the NPE is trained on the simulated datasets, the posterior distribution $p(\theta|x_o)$ can be computed by inputting the observed dataset $x_o$.

A conditional neural density estimator $q_{F(x,\psi)}(\theta)$, utilizing a neural network $F$ and its adjustable network weights $\psi$, is often used as an NPE. In order to train $q_{F(x,\psi)}(\theta)$, the following loss is minimized:

$$\psi^* = \arg\min_{\psi} -\sum_{i=1}^N \log q_{F(x_i,\psi)}(\theta_i), \tag{2}$$

over network weights $\psi$. For a sufficiently expressive $q_F$, $q_{F(x,\psi)}(\theta) \to p(\theta|x)$, as $N \to \infty$.

SNPE aims to improve the accuracy of the approximate posterior for a particular observed dataset $x_o$ iteratively by sampling parameter values from a previous NPE approximation for a given number of rounds. The current NPE approximation is treated as a proposal distribution $\tilde{p}(\theta)$ for the next round. However, training $q_F$ using parameter values drawn from $\tilde{p}(\theta)$ will not converge to the true posterior distribution, but rather to

$$\tilde{p}(\theta|x_o) \propto p(\theta|x_o)\frac{\tilde{p}(\theta)}{p(\theta)}. \tag{3}$$

Many approaches have been developed to overcome this limitation, such as Papamakarios & Murray (2016); Lueckmann et al. (2017); Greenberg et al. (2019). Among all these approaches, we use the automatic

posterior transformation (APT, also known as SNPE-C), as proposed by Greenberg et al. (2019), which has been reported to significantly outperform the others (Lueckmann et al., 2021). For simplicity, we refer to our specific implementation as SNPE hereafter, but we note that other implementations of SNPE can be used with our method.

The general leakage problem can occur when the proposal distribution assigns non-zero probability density outside the support of the prior distribution (Deistler et al., 2022a). In other words, the loss function of SNPE-C (Greenberg et al., 2019) does not force the NCDE to place density within the prior support, causing a mismatch between the posterior and prior support. This can result in extreme values for the proposals, such as generating proposals outside prior bounds, or generating a relatively large number proposals near prior bounds when a suitable transformation is applied. Another possible "leakage" might occur because the neural networks ignore some extreme or invalid data to stabilize the training, leading to unexplored areas of parameter space. This general leakage problem can result in poor performance of SNPE. This is an especially serious problem in situations where leakage is more likely to occur, such as when the training dataset itself contains extreme values of the summary statistics. The truncated SNPE (TSNPE) method (Deistler et al., 2022a) aims to overcome the leakage problem by using a truncated proposal distribution and rejecting proposals if they fall outside of certain quantiles of the prior.

### 2.2.1 Illustrative Example

We consider a sparse vector autoregressive (SVAR) model that has been considered previously in the SBI literature (Thomas et al., 2020; Drovandi et al., 2023). The SVAR model is given by:

$$y_t = X y_{t-1} + \xi_t, \tag{4}$$

where $y_t \in \mathbb{R}^k$ represents the $k$-dimensional observation of the time series at time $t$, $X \in \mathbb{R}^{k \times k}$ is the transition matrix, and $\xi_t \sim \mathcal{N}(0, \sigma^2 \mathbb{I})$ is a $k$-dimensional noise vector with $\sigma^2$ being the noise parameter. The model considers a sparse transition matrix $X$ where the only off-diagonal entries that are non-zero must satisfy the following conditions: if variable $i$ is coupled with variable $j$, then $X_{i,j} \neq 0$ and $X_{j,i} \neq 0$ (note that $X_{i,j}$ is not necessarily equal to $X_{j,i}$) and each variable is coupled to only one other variable. Under this condition, each column will only have one element that is non-zero, and this will also be an off-diagonal element of the matrix. To ensure the stability of the SVAR, the diagonal elements of $X$ are set to -0.1. The parameter space of SVAR can easily scale to higher dimensions by increasing $k$. In this study, the model parameters $\theta \in \mathbb{R}^{k+1}$ are the non-zero off-diagonal entries of $X$ and its variance and we consider $k = 6$, which leads to 7 parameters. This choice is based on the assumption that if SNPE does not produce highly accurate approximations in this low-dimensional case, it is unlikely to be accurate in a higher-dimensional parameter space. We generate an observed dataset of length $T = 1000$ using the true parameter value $\theta = (0.579, -0.143, 0.836, 0.745, -0.660, -0.254, 0.1)$. We use summary statistics to reduce the dimension of the data. Following Thomas et al. (2020); Drovandi et al. (2023), we use the lag 1 autocovariance $\frac{1}{T} \sum_{t=2}^{T} y_t^i y_{t-1}^j$ as the summary statistics, where $y_t^i$ is the $t$th observation of the $i$th time series. We use the sample standard deviation of the $k$ time series to inform $\sigma$. Thus there is a single summary statistic that is intended to be informative about each parameter.

We employ a uniform distribution as the prior, constrained between -1 and 1 for the $k$ parameters and between 0 and 1 for $\sigma$. We denote the generated dataset as $x \in \mathbb{R}^{T \times (k+1)}$ and its corresponding summary as $S(x) \in \mathbb{R}^{(k+1)}$. We find that extreme values of the summary statistics can be produced by parameter values away from the true parameter value. To stabilize the training, we clip simulated datasets with summary statistic outliers (any simulated values greater than 10, around 3% of training datasets). This leads to some regions of the parameter space being unexplored and results in leakage after several rounds of SNPE training. Note that some experimentation was required to obtain a clipping value that led to reasonable results for SNPE.

For illustrative purposes, we run three rounds of SNPE to avoid the leakage problem that occur several rounds after and compare the results with exact posterior samples as the likelihood is easily computable in this example since it is Gaussian. Thus in this case the reference distribution is estimated from exact posterior samples. Ideally, we would expect performance to improve when increasing the number of SNPE

rounds. However, as shown in Figures 1 and 2, even with datasets clipped for every round, SNPE does not improve the accuracy of the estimates as the number of rounds increases.

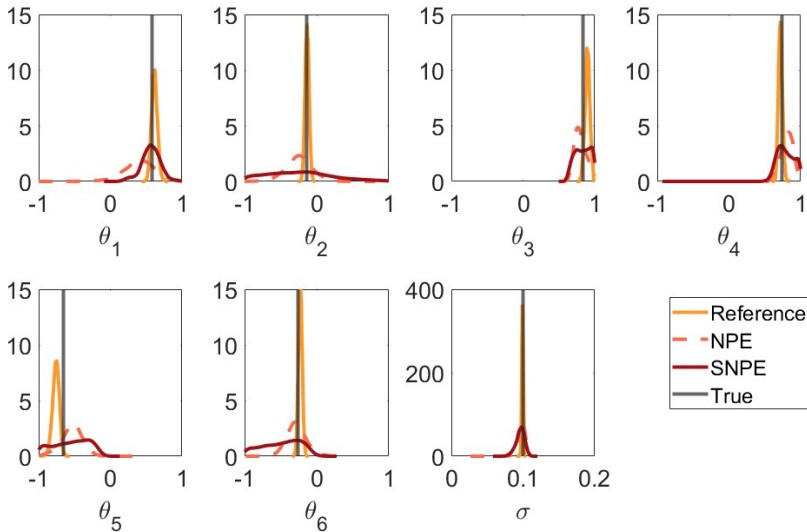

Figure 1: Comparison of marginal posterior distributions between reference distribution (orange), NPE (dashed pink) and SNPE (red), with grey solid lines representing the true values. The SNPE results are based on three rounds.

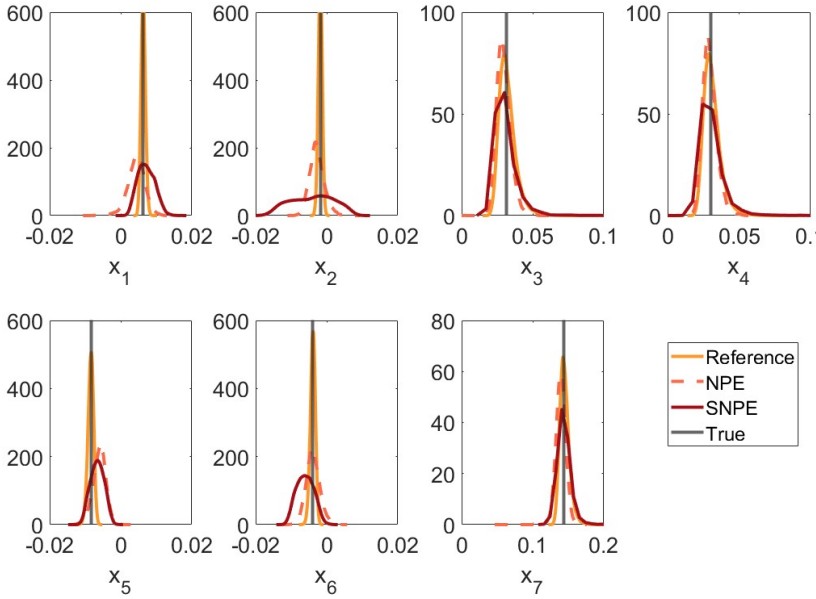

Figure 2: Comparison of posterior predictive distribution of the summary statistics of observation datasets between reference distribution (orange), NPE (dashed pink) and SNPE (red), with grey solid lines representing the true values. The SNPE results are based on three rounds.

## 3   Method

We find that NPE can perform sub-optimally when the prior predictive distribution of the data is complex and has significant variability. When this occurs, the NCDE may not be sufficiently accurate, especially in regions of high posterior support. The sequential version of NPE was originally designed to overcome this issue, however if the initial NPE approximation is not substantially better than the prior, then subsequent rounds of SNPE may suffer from the same issue.

Other approaches to overcome this issue may be to increase the training sample size or to try different configurations of the neural network, but both of these may increase the computational cost substantially and may not address the issue. Instead, we propose the preconditioned NPE (PNPE) method, and its sequential extension below, in order to make NPE methods more reliable.

### 3.1   Preconditioned NPE

For a vague prior distribution $p(\theta)$, parameter values drawn from it might be very far from the true posterior. We suggest using a short run of ABC to refine those parameter values so that they are closer to the true posterior $p(\theta|x_o)$, i.e.,

$$p_\epsilon(\theta|x_o) \propto p(\theta) \int_{\mathbb{R}^D} p(x|\theta)\mathbb{I}(\rho(x_o, x) < \epsilon)dx, \qquad (5)$$

where $\mathbb{I}(\cdot)$ is the indicator function and $\epsilon$ can be chosen considerably larger than what might typically be used in an ABC algorithm. Then we fit a density estimator to those parameter samples. The key idea is to use an efficient ABC algorithm to quickly discard poor regions of the parameter space that generate unusual datasets relative to the observed data, which provides better quality training datasets for NPE. We note that any ABC algorithm could be employed here, but we use the SMC ABC algorithm of Drovandi & Pettitt (2011) in this paper (see Appendix A for a full description of this method). The SMC ABC algorithm generates $n$ samples from a sequence of ABC posteriors based on decreasing ABC thresholds, $\epsilon_1 > \cdots > \epsilon_T$, where $\epsilon_T = \epsilon$ is the target ABC threshold. The sequence of tolerances is determined adaptively, by, at each iteration of SMC, discarding a proportion of the samples, $a \cdot n$, with the highest discrepancy, where $a$ is a tuning parameter. Then, the population of samples is rejuvenated through a resampling and move step. During the move step, a Markov chain Monte Carlo (MCMC) ABC kernel is employed to maintain the distribution of particles based on the current value of the tolerance. The number of MCMC steps $R_t$ to apply to each particle is determined adaptively based on the overall MCMC acceptance rate, that is $R_t = \left\lceil \frac{\log(c)}{\log(1-p_t^{\mathrm{acc}})} \right\rceil$, where $p_t^{\mathrm{acc}}$ is the estimated MCMC acceptance probability at the SMC iteration $t$ and $c$ is a tuning parameter of the algorithm that can be interpreted as the probability that a particle is not moved in the $R_t$ MCMC iterations. A natural stopping rule for the algorithm is when the MCMC acceptance rate becomes intolerably small.

Based on $n$ parameter samples from the ABC posterior, we fit an unconditional normalizing flow $q_G$ (note that other density estimators could be used). Then we can use $q_G$ as the initial importance distribution for the (S)NPE process. We call this melding of ABC and (S)NPE as the preconditioned (S)NPE method. The method is summarized in Algorithm 1.

---

**Algorithm 1** Precondition SNPE

---

1: Choose preconditioning ABC algorithm and SNPE implementation
2: Obtain $\{\theta_i^*\}_{i=1}^n$ from the preconditioning ABC algorithm
3: Set $\phi^* \leftarrow \arg\min\limits_{\phi} \sum\limits_{i=1}^{n} -\log q_{G(\phi)}(\theta_i^*)$
4: Perform SNPE using initial importance distribution $q_{G(\phi^*)}(\theta)$

---

If we obtain a well-trained unconditional normalizing flow, this unconditional normalizing flow can act as the initial importance distribution, i.e., $\tilde{p}(\theta)$ in Equation 3, and to draw samples for training the NPE. Following Papamakarios & Murray (2016), it is noted that given an expressive enough conditional normalizing flow,

NPE converges to the true posterior $p(\theta|x_o)$ as $N \to \infty$, with an appropriate importance re-weight if $\tilde{p}(\theta) \neq p(\theta)$.

Choosing a suitable value of $\epsilon$ for our method requires some thought. A smaller value of $\epsilon$ will focus in on more promising regions of the parameter space, but will increase the computational time of the preconditioning step. A larger value of $\epsilon$ will lead to a fast preconditioning step, but may not eliminate enough of the poor parts of the space to improve the training of the NCDE. In this paper we use an MCMC acceptance rate of 10% (unless otherwise specified) as the stopping criteria for the SMC ABC algorithm in the preconditioning step. For our examples we find that this choice is effective at balancing the aforementioned objectives. We note that other choices are possible.

Furthermore, once these poor simulations have been removed, we find NPE to be more effective than ABC, since ABC requires an exponentially increasing number of simulations to drive $\epsilon$ to 0. To avoid the scaling problem of ABC, the preconditioning step only takes a short run of ABC, and thus we are not interested in driving $\epsilon$ to 0.

## 3.2 Computational cost

We now consider computational cost for P(S)NPE and compare it with SNPE. The preconditioning step can be considered as the initial round of NPE where the total number of simulated datasets generated during SMC ABC is denoted as $n_{\text{ABC}}$. Hence, it is worth noting that P(S)NPE, like SNPE, is not amortized since it requires running an ABC algorithm for each observation datasets $x_0$.

Furthermore, for complex real-world problems, the simulation time may depend on the parameter values, and parameter values with very low posterior support can produce substantially longer simulation times. For such problems, it is important from a computational perspective to quickly eliminate such regions from the parameter space, as is the motivation of the preconditioning ABC algorithm. Thus, for problems where SNPE does not perform well, we find PSNPE to be substantially more computationally efficient in terms of compute time.

We perform an analysis of the trade-off between computational cost and estimation accuracy by using different choices for the % of stopping rule for the preconditioning step in the SVAR example. For reproducibility, we use 20 different random seed values for each choice of the stopping rule. We compute and record the average values we choose for the % and its corresponding total number of simulations in Table 1.

| MCMC ACCEPTANCE RATE | 15% | 12% | 10% | 8% | 5% |
|---|---|---|---|---|---|
| SIMULATIONS IN ABC | 15603 | 33585 | 62402 | 100398 | 244667 |

Table 1: **Computational cost for preconditioning step:** The first row of the table refers to the % stopping rule we selected to obtain samples from ABC, while the second row indicates the average of total number of simulations ABC used for the corresponding %.

To investigate the performance of this trade-off, we use maximum mean discrepancy (MMD) as a metric to compare the approximate distribution from ABC with the reference distribution. We also run NPE with the same number of simulations as ABC and compute its MMD. In Figure 3, we plot the total number of simulations versus MMD for ABC, NPE and PNPE. The solid lines represent the average MMD values based on 20 MMD values from 20 different seed values. The number of simulations for PNPE includes an additional 10k simulations generated after the corresponding preconditioning step. It is surprising that with an increasing number of simulations, the accuracy of NPE can vary significantly. For some random seed values, the accuracy of NPE even decreases when using more than approximately 70,000 model simulations. Compared with ABC and PNPE, which have narrow uncertainty intervals, the accuracy of NPE seems to highly depend on the random seed value.

We record that $3-5\%$ of training datasets are clipped before training NPE, which means around 2,100-3,500 simulated datasets were ignored, and the corresponding parameter space is not explored well. This might indicate the reason why the accuracy of NPE might reduce under a large number of simulations for some

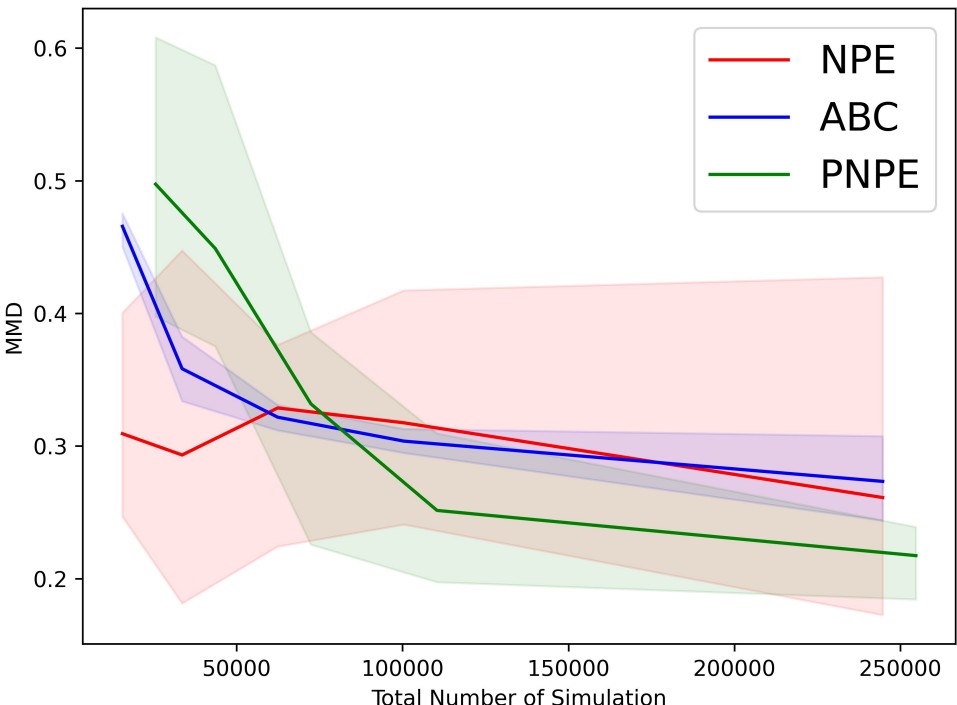

Figure 3: **Computational cost versus MMD for ABC, NPE and PNPE for SVAR example:** Comparison of MMD values for ABC (blue), NPE (red) and PNPE (green) with corresponding color bands indicating the 95% uncertainty interval. The x-axis shows an increasing number of simulations required for each algorithm.

random seed values. It is evident that an increasing number of simulations can cause a decrease in MMD values for both ABC and PNPE. However, when a large number of simulations are used in the preconditioning step, the following NPE step does not provide a large improvement in accuracy. The small improvement in accuracy may not be worth the cost incurred by the large number of model simulations. Having a percentage too large will stop the ABC too early, and we will lose the benefit of the preconditioned step. Having a percentage too small will result in ABC running too long and using too many model simulations.

### 3.3 Illustrative Example Revisited

We apply PNPE to the illustrative example shown in Section 2.2.1. For the preconditioning step, we use the adaptive SMC ABC algorithm proposed by Drovandi & Pettitt (2011), with tuning parameters $n = 1k$, $a = 0.5$, and $c = 0.01$. We employ an unconditional normalizing flow as the unconditional density estimator. For this, we use the state-of-the-art neural spline flow implemented in the `Pyro` package.

In order to make a fair comparison between P(S)NPE and SNPE, we use the same number of simulations as in the SMC ABC algorithm, denoted $n_{\text{ABC}}$, to train the initial round of NPE. We run the SMC ABC algorithm ten times to obtain the average number of simulations it requires, which is $n_{\text{ABC}} = 54k$. For illustrative purposes, we only run two rounds of SNPE and compare it with PNPE.

The estimated marginal posterior plots are displayed in Figure 4, where the black solid lines represent the true parameter values. It is evident that our PNPE method (in a single round) produces a substantially sharper approximation of the posterior compared to that of SNPE.

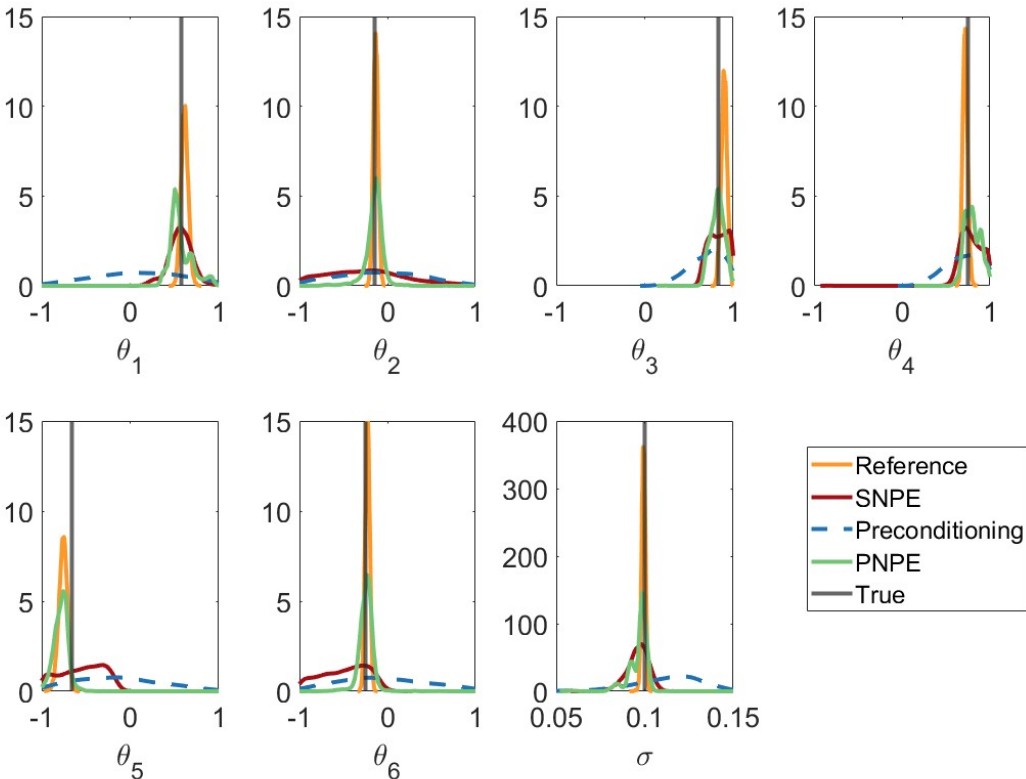

Figure 4: **Performance on SVAR model.** Comparison of marginal posterior distributions between reference distribution (orange), SNPE (red), preconditioning step (blue dash) and PNPE (green solid), with grey solid lines representing the true values.

It is evident that even a short run of the ABC algorithm gives a reasonable posterior approximation to train an unconditional normalizing flow as the unconditional density estimator, which then generates samples for NPE training. Figures 4 and 5 show that the improved parameter posteriors leads to more accurate posterior predictive distributions of the summaries compared to SNPE. This indicates that with a good starting point, NPE can further improve accuracy.

In Appendix C.1, we show the results for 10 different datasets, each with 10 different random seeds for reproducibility purposes. We use the same stopping rule for the preconditioning step across all datasets. It is evident that the performance between ABC and NPE is close under the same number of simulations. In some cases, NPE performs better than ABC as it has sufficient data to learn while not too much parameter space is ignored. However, PNPE performs better than SNPE as the leakage problem occurs. We also compare NPE and PNPE with the same number of simulations and find that the performance of PNPE is still better than NPE.

## 4  Further Experiments

We present five benchmarking tasks (two from Lueckmann et al. (2021) and three from the SBI literature) and two additional examples, including our motivating example, where SNPE, perhaps surprisingly, does not produce highly accurate posterior distributions. To fairly compare our method with vanilla SNPE and potentially TSNPE (if the leakage issue is encountered), we run the ABC algorithm 10 times and compute the average total number of simulations that the ABC algorithm requires. We then use this same number of

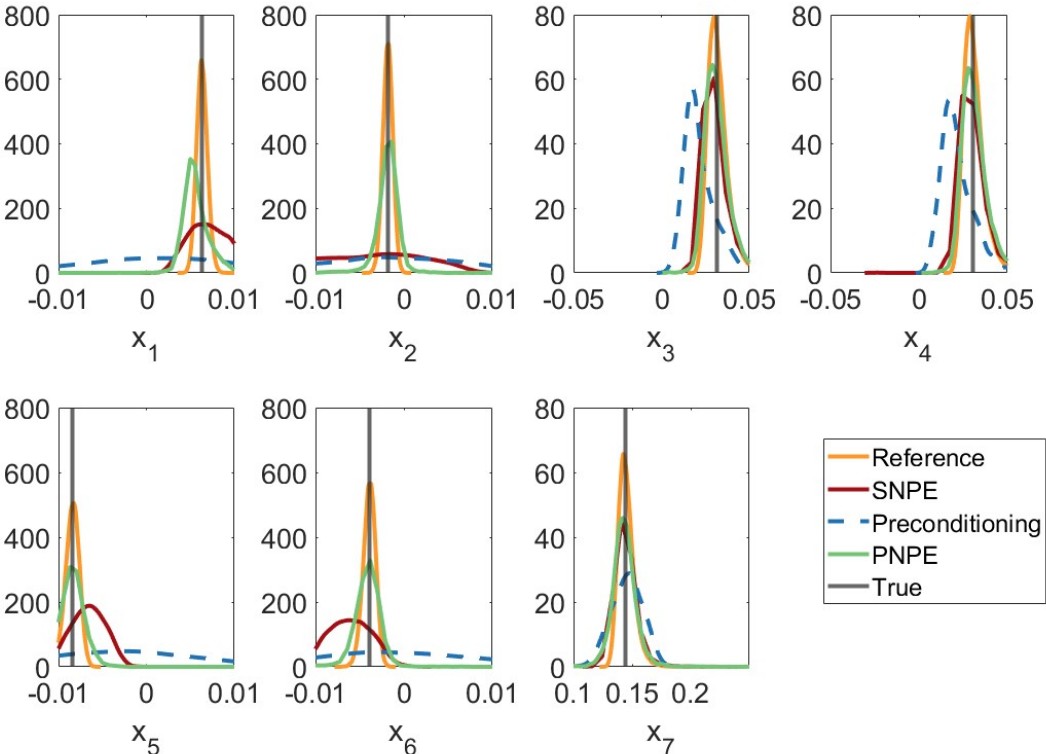

Figure 5: **Performance on SVAR model.** Comparison of posterior predictive distributions of the summary statistics of observation datasets between reference distribution (orange), SNPE (red), preconditioning step (blue dash) and PNPE (green solid), with grey solid lines representing the true values.

simulations for the initial NPE in both SNPE and TSNPE. For all experiments, we utilize the adaptive SMC ABC algorithm proposed by Drovandi & Pettitt (2011) for the preconditioning part with tuning parameters $n = 1k$, $a = 0.5$, $c = 0.01$, and use an unconditional normalizing flows as the unconditional density estimator. For SNPE, we use 10k samples for each round of training.

We find that even with a single round of PNPE, there can be a significant improvement in performance. The code to reproduce the results has been included as supplementary material.

## 4.1 Benchmarking example

We compare PSNPE and SNPE across five popular benchmarking simulators (see Appendix B.1 for a detailed description). We use the `sbibm` package (Lueckmann et al., 2021) for the two-moon and SLCP simulators, and the `ELFI` package (Lintusaari et al., 2018) for the other benchmark simulators. Since the ground-truth posteriors are available, we can use specific performance metrics for comparison. We use a classifier two-sample test (C2ST), where a score of 0.5 indicates that the approximate posterior is indistinguishable from the true posterior, and a score of 1 signifies that the classifier can completely separate the approximate posterior from the true posterior. Additionally, we use the maximum mean discrepancy (MMD) between ground truth posterior and approximate posterior to quantify performance. We refrain from using the negative log probability of true parameters as a performance metric because our method is not amortized. It is evident from Table 2 that SNPE and PSNPE achieve similar results, except for the g-and-k example, the results of which we describe in more detail next.

| MODEL | C2ST | MMD |
|---|---|---|
| TWO MOON | **0.527**\0.528 | **0.00067**\0.0007 |
| SLCP | **0.664**\0.691 | **0.00065**\0.001 |
| MA(2) | 0.887 \**0.856** | 0.104 \**0.0966** |
| G-AND-K | 0.975 \**0.727** | 0.553 \**0.311** |
| TOAD EXAMPLE | 0.899 \**0.887** | 0.120 \**0.111** |

Table 2: **Performance on popular benchmarking simulators.** Classification accuracy (CS2T) and maximum mean discrepancy (MMD) computed for SNPE\PSNPE with same number of simulation for ABC and NPE followed by 2 rounds SNPE, each round with $N = 10k$ simulations. The bold indicates a better performance metric value.

For the benchmark examples in Table 2 where the performance of SNPE and PSNPE are similar, the initial training dataset is not extreme and the NCDE fits reasonably well. In more realistic examples, the prior predictive distribution of the summaries may be much more complex and contain extreme values. In simple examples, we can create more complex prior predictive distributions by widening the prior. Here we reconsider the g-and-k benchmark example but widen the prior distribution.

We describe the details of the g-and-k model and its prior distribution in Appendix B.1. We use full datasets instead summary statistics for both PNPE and SNPE. For visualization purposes, we plot the marginal approximate posterior distribution for each method of the estimated results. It is evident that ABC performs better than NPE under the same number of simulations (Figure 6a), where the ABC posterior is able to concentrate on the true value of $B$ and $k$. As shown in Figure 6b, PNPE performs better than SNPE.

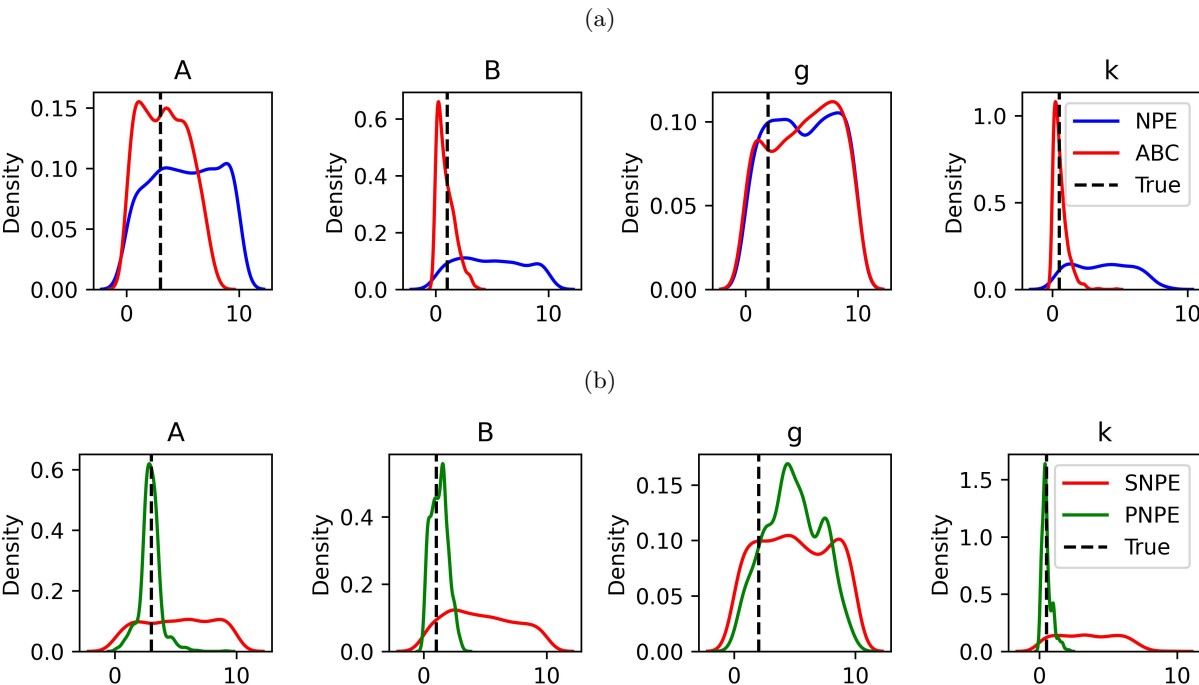

Figure 6: **Performance on G-and-K example:** (a) Comparison of marginal posterior distributions between the ABC preconditioning step (red) and NPE (blue), with black dashed lines representing the true values; (b) Comparison of marginal posterior distributions between PNPE (green) and SNPE (red), with black dashed lines representing the true values. The result of SNPE uses 2 rounds with same number of simulation as the ABC preconditioning step.

### 4.2 High-dimensional SVAR model

To investigate how our method scales to higher dimensional problems, we take the illustrative SVAR example from before and consider $k = 20$, which leads to 21 parameters. We detail the experimental settings in Section B.2. To ensure a fair comparison, we run the SMC ABC algorithm ten times using a 10% acceptance rate as the stopping criterion and calculate the average number of simulations it takes. We then use the same number of simulations, approximately $n_{\text{ABC}} \approx 45k$, to train the initial NPE. Hence, the total number of simulations for both PNPE and SNPE is the same (55k in total). To stabilize the training, we apply the same clipping technique used in the previous low-dimensional case, which results in approximately 11% of the training samples being removed in the initial round of NPE and about 1% to 2% in the second rounds. Starting from the third round, SNPE is unable to sample any parameter values from the neural networks due to a severe leakage issue (Deistler et al., 2022a).

The estimated marginal posterior plots are displayed in Figure 7, where the black solid lines represent the true parameter values. It is evident that as the number of parameter dimensions increases, training the unconditional normalizing flows becomes more challenging. With well-trained unconditional normalizing flows, PNPE outperforms SNPE in high-dimensional cases (in this example, except for parameter $\theta_3$.).

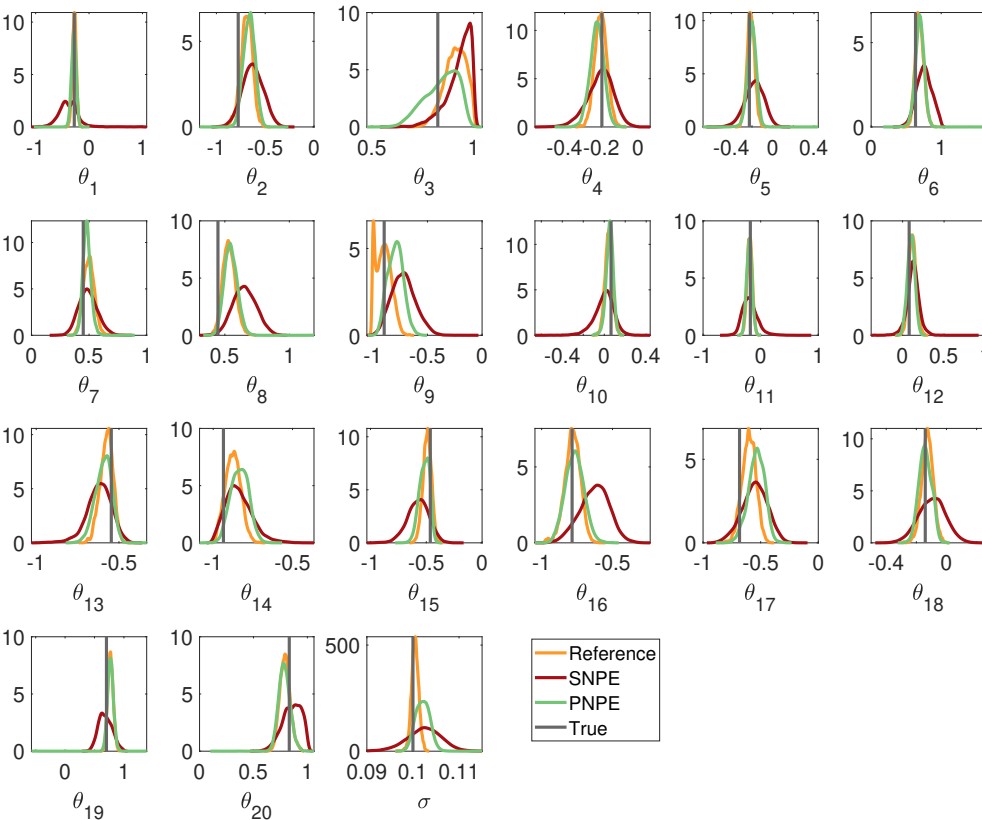

Figure 7: **Performance on SVAR model with 21 parameters.** Comparison of marginal posterior distributions between reference distribution (orange), SNPE (red) and PNPE (green), with grey solid lines representing the true values. The result of SNPE uses 2 rounds.

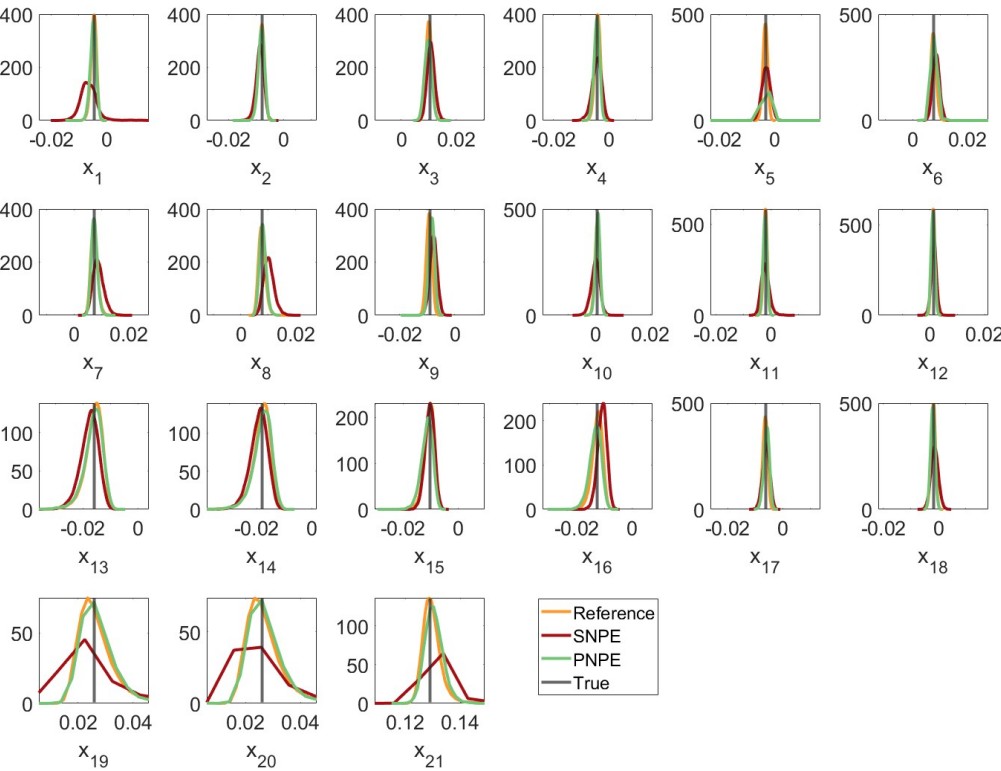

Figure 8: **Performance on SVAR model with 21 parameters.** Comparison of posterior predictive distributions of the summary statistics of observation datasets between reference distribution (orange), SNPE (red) and PNPE (green), with grey solid lines representing the true values. The result of SNPE uses 2 rounds.

### 4.3 Biphasic Voronoi cell-based model

Finally, we consider a challenging real-world problem in cancer biology: calibrating the biphasic Voronoi cell-based model (BVCBM) (Wang et al., 2024) that models tumor growth. The model uses a parameter $\tau$ to divide the tumor into two growth phases. Here, the term 'growth phase' refers to the different growth patterns of the tumor. There are four parameters that govern tumor growth during each phase, namely $(p_0, p_{psc}, d_{max}, g_{age})$, where $p_0$ and $p_{psc}$ are the probability of cell proliferation and invasion, respectively, $d_{max}$ is the maximum distance between cell and nutrient, and $g_{age}$ is the time taken for a cell to be able to divide. Thus there are nine parameters in total, four parameters each of two phases, and the parameter $\tau$ at which the growth phase changes. In this paper, we calibrate to two real-world pancreatic cancer datasets Wade (2019), which describe tumor growth as time series data. The datasets span 26 and 32 days, respectively, with measurements taken each day. While the ground truth posteriors are unknown for those datasets, we compute posterior predictive distributions to assess if SNPE and PNPE can effectively calibrate the model to the data.

We employ vague prior distributions for all parameters. Specifically, we use a Uniform distribution constrained between 1 and 24 hours $\times$ the number of days for $g_{age}$ during both growth phases. Additionally, we use a Uniform distribution constrained between 2 and the number of days minus 1 for $\tau$. The prior distributions for the remaining parameters are detailed in Appendix B.3.

As reported by Wang et al. (2024), CPU times for model simulation range from 1.76 to 137.27 seconds per simulation when using samples from the prior distribution. This implies that 10k simulations for the first round of SNPE take approximately 2 hours. Consequently, the initial stages of SNPE are computationally expensive. In contrast, the ABC preconditioning step takes around 10-15 minutes. This is due to the fact that the longer simulation times tend to also lead to large discrepancies with the observed data, and such samples are quickly rejected by ABC. For the ABC part, around 18k and 16k simulations are used for the 26-day and 32-day pancreatic cancer datasets, respectively.

We observed a leakage problem occurring in rounds 8 and 5 for the 26-day and 32-day datasets, respectively. Hence, we perform 10 rounds of TSNPE for all three datasets. At round 10, the acceptance rate of SNPE for the 26-day measurement dataset is above 50%, leading us to utilize rejection sampling for sample generation. While a 50% acceptance rate for rejection sampling is acceptable, it is less efficient compared to direct simulation from the trained conditional normalizing flows in the previous round. Hence, we employ TSNPE for this dataset. For the 32-day measurement dataset, only 0.000% of samples are accepted at round 8, making it computationally expensive. Consequently, we use TSNPE for this dataset.

To estimate the posterior predictive distributions, we sample 1k parameter values from (T)SNPE and PNPE, using them to simulate datasets. We then plot these data in the form of credible intervals. As a baseline, we show the prior predictive distributions in Appendix C.2. The top and bottom rows of Figure 9 displays the posterior predictive distribution for the 26-day and 32-day datasets, respectively (the same plots but on the log scale are shown in Appendix C.2). It is evident that both SNPE and TSNPE provide biased estimations, as the posterior predictive does not capture the observed data well. The posterior predictive distribution for the preconditioning step (middle column) shows that the ABC step can capture the data reasonably well since the observed data lie within the 90% posterior predictive interval. Our method (third column) provides a better fit, as the variance of the posterior predictive distribution is tighter than that of the preconditioning step and still captures the observed data. This demonstrates that even one round of PNPE can perform more accurate estimations based on our results.

## 5 Discussion

We present a neural SBI method that is both simple and easy to deploy, designed to enhance the accuracy of SNPE methods. Our method, termed preconditioned neural posterior estimation (PNPE) and its sequential version, PSNPE, employs an ABC algorithm for the initial step. This algorithm is used to efficiently filter out poor regions of the parameter space. Additionally, we use the ABC posterior samples to train an unconditional density estimator $q_G$, enabling $q_G$ to serve as the initial proposal distribution for SNPE. The core concept is that an improved starting point can significantly enhance the accuracy of SNPE estimations. Indeed, we obtained very good results with PNPE. Lueckmann et al. (2017); Deistler et al. (2022b) have proposed similar ideas to improve accuracy of SNPE. They train a classifier to predict samples that fall into rejection criterion based on a certain distance metric. However, such a classifier can falsely reject samples. Our method can avoid this problem and hence is more principled compared to their method.

We showcase several examples where either SNPE failed to perform inference effectively, such as in the SVAR case, or produced biased results, as observed in the BVCBM. For the SVAR example, SNPE methods struggle due to the impact of low-quality samples from certain parameter space regions, adversely affecting the training process. The ABC method can efficiently eliminate these bad samples, thereby enhancing the training. For cases where SNPE results in biased estimations, our methods were effective at accurately fitting observed data (real data for BVCBM example). This is substantiated by our empirical results for the posterior predictive distribution discussed in the experimental section.

Although our method demonstrates the capability to enhance estimation accuracy, it does have some limitations. Firstly, our method requires model simulations in the ABC preconditioning step, which may lead to greater computational demands in situations where SNPE methods perform well. However, by performing the preconditioning step, significantly fewer model simulations may be required in the SNPE part to achieve high accuracy. In this paper we used SMC ABC for the preconditioning step, but we note that other ABC algorithms or SBI methods could be used. We do not recycle the simulations performed in the ABC preconditioning step for the SNPE phase, but it could be possible to modify our method to exploit

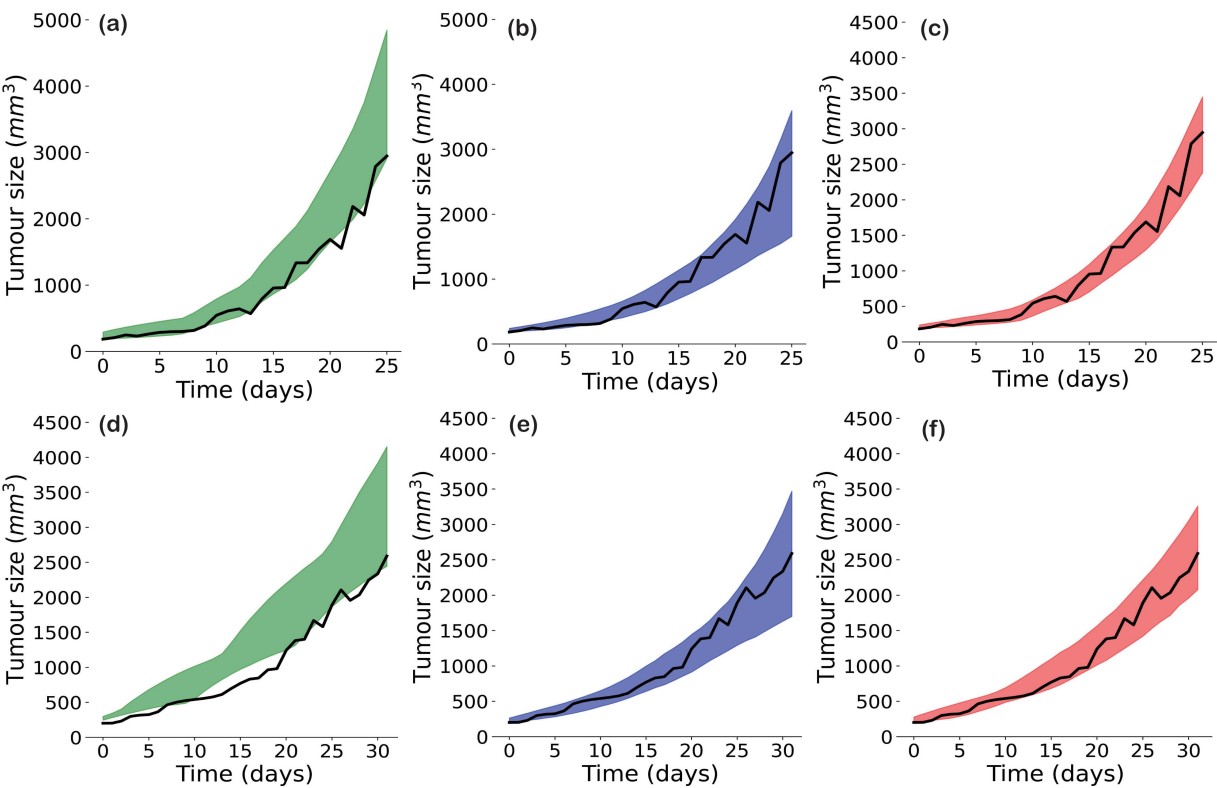

Figure 9: **Posterior predictive distributions for two pancreatic cancer datasets.** 90% Posterior predictive interval plots for SNPE (**a**), preconditioning step (**b**) and PNPE (**c**) for the 26-day dataset; 90% Posterior predictive interval plots for SNPE (**d**), preconditioning step (**e**) and PNPE (**f**) for the 32-day dataset.

these model simulations. Here we used the acceptance rate as the stopping criterion for the preconditioning step, but another option could be to check after each iteration of SMC ABC and stop the preconditioning if a suitable NCDE is found. Secondly, the choice of an unconditional density estimator necessitates careful consideration. In scenarios involving low-dimensional parameter spaces, a kernel density estimator might be a preferable option compared to unconditional normalizing flows.

It is important to note that the preconditioning step of our method requires manual selection of the summary statistics or discrepancy function. However, this choice is not as critical as in a typical ABC application, since we aim to remove poor parts of the parameter space rather than achieve a highly accurate posterior approximation. If it is difficult to manually select summary statistics, the ABC preconditioning step could use discrepancy functions that do not rely on summary statistics such as Wasserstein ABC (Bernton et al., 2019) and K2-ABC (Park et al., 2016), followed by an NPE method that incorporates a summary network (Jiang et al., 2017). This approach would have the benefit that the training of the summary network will not be adversely affected by extreme simulated data. In this paper we used the same summary statistics as in the preconditioning step for the subsequent rounds of NPE. However, we note it would be possible to use different summaries after the preconditioning step, and possibly use automated summary statistic selection methods (Fearnhead & Prangle, 2012; Chen et al., 2023).

In this paper we considered the well-specified scenario, where the model is either known to be correct or can provide a good fit to the data with a suitable choice of parameter values. Standard neural SBI methods are known to potentially perform poorly under model misspecification Bon et al. (2023); Cannon et al. (2022); Schmitt et al. (2023). Our preconditioning method may be useful in the misspecified scenario, since ABC

are known to perform reasonably well under model misspecification. That is, ABC still converges onto the pseudo-true parameter value Frazier et al. (2020). The preconditioning step could be followed by a robust neural SBI method such as Kelly et al. (2024); Huang et al. (2024); Gloeckler et al. (2023); Ward et al. (2022). We plan to explore this in future research.

Overall, PNPE employs a preconditioning step to focus on important parts of the parameter space, thereby creating a good starting point for training SNPE and enhancing estimation accuracy. We have empirically demonstrated that PNPE is capable of producing more accurate estimations in complex real-world problems.

## Acknowledgement

We thank the computational resources provided by QUT's High Performance Computing and Research Support Group (HPC). Xiaoyu Wang, Ryan P. Kelly and Christopher Drovandi were supported by an Australian Research Council Future Fellowship (FT210100260).

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

# A   Further background for SMC ABC

We provide a detailed description of the adaptive SMC ABC algorithm we used in this paper and provide pseudocode in Algorithm 2 for reference. This algorithm starts by drawing $N$ independent samples from the prior distribution $p(\theta)$, represented as $\{\theta_i\}_{i=1}^N$. For each sample $\theta_i$ (known as a particle), the algorithm simulates a dataset $x_i$ from the stochastic model and calculates the corresponding discrepancy $\rho_i = \rho(x_i, x_o)$, resulting in the pair set $\{\theta_i, \rho_i\}_{i=1}^N$. These pairs are then arranged in order of increasing discrepancy such that $\rho_1 < \rho_2 < \cdots < \rho_N$. The first tolerance threshold, $\epsilon_1$, is set as the largest discrepancy, $\rho_N$. To move through the target distributions, the algorithm adjusts the tolerance dynamically. The next tolerance, $\epsilon_t$, is set as $\rho_{N-N_a}$, where $N_a = \lfloor Na \rfloor$, and $a$ is a tuning parameter. Essentially, in each step, the algorithm discards the top $a \times 100\%$ of particles with the highest discrepancies. After discarding these particles, only $N - N_a$ particles remain. To replenish the set back to $N$ particles, the algorithm resamples $N_a$ times from the 'alive' particles, copying both the parameter and discrepancy values. This process, however, leads to duplicates in the particle set. To add variety to the set, the algorithm applies an MCMC ABC kernel to each resampled particle. The parameters for the MCMC proposal distribution $q_t(\cdot|\cdot)$ are derived from the current particle set. For instance, if using a multivariate normal random walk proposal, its covariance $\Sigma_t$ is based on the particle set's sample covariance. The acceptance of a proposed parameter (assuming a symmetric proposal) and simulated data is determined by the equation:

$$p_t = \min\left(1, \frac{p(\tilde{\theta})}{p(\theta)}\mathbb{I}(\rho(x_o, \tilde{x}) < \epsilon_t)\right), \tag{6}$$

where $\tilde{\theta} \sim q(\cdot|\theta)$ and $\tilde{x} \sim p(\cdot|\tilde{\theta})$ are proposed parameter values and dataset, respectively. However, proposals may be rejected, leaving some particles unchanged. To address this, the algorithm performs $R_t$ iterations of the MCMC kernel on each particle, where $R_t = \left\lceil \frac{\log(c)}{\log(1-p_t^{\mathrm{acc}})} \right\rceil$, where $c$ is a tuning parameter of the algorithm that can be interpreted as the probability that a particle is not moved in the $R_t$ iterations. The acceptance probability $p_t^{\mathrm{acc}}$ is estimated from trial MCMC ABC iterations and used to compute $R_t$ for the next set of MCMC ABC iterations. For this adaptive SMC ABC algorithm, two stopping rules can be used. The first stopping rule halts the ABC algorithm when the maximum discrepancy is below a set tolerance, $\epsilon_T$. The second stopping rule terminates the algorithm when the MCMC acceptance probability $p_t^{\mathrm{acc}}$ falls below a predefined threshold $p_{\mathrm{acc}}$. Here we choose the second rule, and since we only require a short run of ABC, we set $p_{\mathrm{acc}}$ to be higher than what is typically used in an ABC analysis.

---

**Algorithm 2** Adaptive SMC ABC

---

**Input:** The observed data $x_o$, the stochastic model $p(x|\theta)$, distance function $\rho(\cdot, \cdot)$, prior distribution $p(\theta)$, number of particles $N$, tuning parameters $a$ and $c$ for adaptive selection of discrepancy thresholds and selecting the number of MCMC iterations in the move steps, target tolerance $\epsilon_T$, initial number of trial MCMC iterations $S_{\text{init}}$, minimum acceptable MCMC acceptance rate $p_{\min}$

**for** $i = 1, \ldots, N$ **do**
    Simulate $x_i \sim p(x|\theta_i)$ where $\theta_i \sim p(\theta)$
    Compute $\rho_i = \rho(x_o, x_i)$
**end for**
Sort $\{\theta_i\}_{i=1}^N$ by $\{\rho_i\}_{i=1}^N$ such that $\rho_1 \leq \rho_2 \leq \cdots \leq \rho_N$
Set $N_a = \lfloor aN \rfloor$, $\quad t = 2$, $\quad \epsilon_t = \rho_{N-N_a}$, $\quad \epsilon_1 = \rho_N$, $\quad S_t = S_{\text{init}}$, $\quad \tilde{p}_t^{\text{acc}} = 1$
**while** $\epsilon_{t-1} > \epsilon_T$ **or** $\tilde{p}_t^{\text{acc}} > p_{\min}$ **do**
    Compute $\Sigma$ as the sample covariance matrix of $\{\theta_i\}_{i=1}^{N-N_a}$
    Generate $\{\theta_i\}_{i=N-N_a+1}^N$ by resampling from $\{\theta_i\}_{i=1}^{N-N_a}$ with replacement
    **for** $i = N - N_a + 1, \ldots, N$ **do**
        **for** $j = 1, \ldots, S_t$ **do**
            Simulate $\tilde{x} \sim p(x|\tilde{\theta})$ based on proposal $\tilde{\theta} \sim \mathcal{N}(\theta_i, \Sigma)$
            Compute $\tilde{\rho} = \rho(x_o, \tilde{x})$
            Compute $p_t^{i,j} = \min\left(1, \frac{p(\tilde{\theta})}{p(\theta)}\mathbb{I}(\tilde{\rho} < \epsilon_t)\right).$
            With probability $p_t^{i,j}$, set $\theta_i = \tilde{\theta}$ and $\rho_i = \tilde{\rho}$; otherwise, retain the current values of $\theta_i$ and $\rho_i$
        **end for**
    **end for**
    $\tilde{p}_t = \sum_{i=N-N_a+1}^N \sum_{j=1}^{S_t} p_t^{i,j} / \left(S_t(N - N_a)\right)$
    $R_t = \lceil \log(c) / \left(1 + \log(1 - \tilde{p}_t)\right) \rceil$
    **for** $i = N - N_a + 1, \ldots, N$ **do**
        **for** $j = R_t - S_t, \ldots, R_t$ **do**
            Simulate $\tilde{x} \sim p(x|\tilde{\theta})$ based on proposal $\tilde{\theta} \sim \mathcal{N}(\theta_i, \Sigma)$
            Compute $\tilde{\rho} = \rho(x_o, \tilde{x})$
            Compute $p_t^{i,j} = \min\left(1, \frac{p(\tilde{\theta})}{p(\theta)}\mathbb{I}(\tilde{\rho} < \epsilon_t)\right).$
            With probability $p_t^{i,j}$, set $\theta_i = \tilde{\theta}$ and $\rho_i = \tilde{\rho}$; otherwise, retain the current values of $\theta_i$ and $\rho_i$
        **end for**
    **end for**
    $\tilde{p}_t^{\text{acc}} = \sum_{i=N-N_a+1}^N \sum_{j=1}^{R_t} p_t^{i,j} / \left(R_t(N - N_a)\right)$
    $S_{t+1} = \lceil R_t/2 \rceil$
    Sort $\{\theta_i\}_{i=1}^N$ by $\{\rho_i\}_{i=1}^N$ such that $\rho_1 \leq \rho_2 \leq \cdots \leq \rho_N$
    Set $\epsilon_{t+1} = \rho_{N-N_a}$, $\quad \epsilon_t = \rho_N$
    $t = t + 1$
**end while**
**return** Samples $\{\theta_i\}_{i=1}^N$ from ABC posterior

---

# B    Experimental Details

We use the adaptive SMC ABC algorithm Drovandi & Pettitt (2011) in the preconditioning step for all experiments. We set the tuning parameters as $a = 0.5$, $c = 0.01$ and use 1k particles for the algorithm. As the stopping rule, we set the target MCMC acceptance rate at 10%, unless otherwise specified. For the unconditional density estimator, we employ unconditional normalizing flows using the `Pyro` package Bingham et al. (2019), with a spline coupling layer using the transformation:

$$Y_{1:d} = g_{\tilde{\theta}}(X_{1:d}) \tag{7}$$

$$Y_{(d+1):D} = h_\phi(X_{(d+1):D}; X_{1:d}) \tag{8}$$

where $\mathbf{X}$ are the inputs, $\mathbf{Y}$ are the outputs, e.g., $X_{1:d}$ represents the first $d$ elements of the inputs, $g_{\tilde{\theta}}$ is either the identity function or an elementwise rational monotonic spline with parameters $\tilde{\theta}$, and $h_\phi$, where $\phi$ is element-wise bijection parameter, is a conditional elementwise spline, conditioning on the first $d$ elements. Regarding the neural networks, we use four fully-connected layers and set the count bins to 16. Furthermore, if the dimensions of the parameter space are less than three, indicating a low-dimensional case, we also consider kernel density estimation with a Gaussian kernel as the unconditional density estimator, as implemented in the `Scikit-learn` package Pedregosa et al. (2011). For APT and TSNPE, we use the implementation of the `sbi` package Tejero-Cantero et al. (2020) with default settings.

For SNPE and TSNPE, we use conditional neural spline flows Durkan et al. (2019). We use five coupling layers, with each coupling layer using a multilayer perceptron of two layers with 50 hidden units. The flow is trained using the Adam optimizer with a learning rate of $5 \times 10^{-4}$ and a batch size of 256. Flow training is stopped when either the validation loss, calculated on 10% of the samples, has not improved over 50 epochs or when the limit of 500 epochs is reached.

## B.1    Benchmarking examples

A number of popular benchmarking models in the SBI literature exist where the ground truth posteriors are available. For the two-moon model and the SLCP model, we follow the model specifications in Lueckmann et al. (2021). For the rest of the models, we refer to Lintusaari et al. (2018). After the preconditioning step, we generate 10k simulations from the unconditional normalizing flows and pair them with their simulated data to process SNPE.

### B.1.1    Two moons model

The two moons model exhibits both global (bimodality) and local (crescent shape) structures in the posterior. For the preconditioning step, the total number of model simulations is around 30k. Since $\boldsymbol{\theta}$ is low-dimensional in this model, we use KDE as the unconditional density estimator and achieve results similar to those obtained with unconditional normalizing flows.

| | |
|---|---|
| **Simulator** | $\boldsymbol{x}\lvert\boldsymbol{\theta} = \begin{bmatrix} r\cos(\alpha) + 0.25 \\ r\sin(\alpha) \end{bmatrix} + \begin{bmatrix} -\lvert\theta_1 + \theta_2\rvert/\sqrt{2} \\ -(\theta_1 - \theta_2)/\sqrt{2} \end{bmatrix}$, where $\alpha \sim U(-\pi/2, \pi/2)$, $r \sim \mathcal{N}(0.1, 0.01^2)$ |
| **Prior** | $\boldsymbol{\theta} = (\theta_1, \theta_2)$, $\quad \theta_i \sim U(-1, 1)$ for $i = 1, 2$ |
| **Dimensionality** | $\boldsymbol{\theta} \in \mathbb{R}^2, \boldsymbol{x} \in \mathbb{R}^2$ |
| **References** | Greenberg et al. (2019) |

### B.1.2    SLCP with Distractors model

The SLCP model is designed to have a simple likelihood and a complex posterior, with uninformative dimensions (distractors) added to the observations. The preconditioning step uses around 35k model simulations.

| | |
|---|---|
| **Simulator** | $\boldsymbol{x} = (\boldsymbol{x}_1, \ldots, \boldsymbol{x}_{100}), \quad \boldsymbol{x} = p(\boldsymbol{y}),$ |

where $p$ re-orders the dimensions of $\boldsymbol{y}$ with a fixed random permutation;

$$\boldsymbol{y}_{[1:8]} \sim \mathcal{N}(\boldsymbol{m_\theta}, \boldsymbol{S_\theta}), \quad \boldsymbol{y}_{[9:100]} \sim \frac{1}{20} \sum_{i=1}^{20} t_2(\boldsymbol{\mu}^i, \boldsymbol{\Sigma}^i),$$

where $\boldsymbol{m_\theta} = \begin{bmatrix} \theta_1 \\ \theta_2 \end{bmatrix}, \quad \boldsymbol{S_\theta} = \begin{bmatrix} s_1^2 & \rho s_1 s_2 \\ \rho s_1 s_2 & s_2^2 \end{bmatrix}, \quad s_1 = \theta_3^2, \quad s_2 = \theta_4^2, \quad \rho = \tanh(\theta_5),$

$\boldsymbol{\mu}^i \sim \mathcal{N}(0, 15^2 \mathbf{I}), \quad t_2$ is student t-distribution with degree of freedom 2,

$\Sigma_{j,k}^i \sim \mathcal{N}(0, 9),$ for $j > k, \quad \Sigma_{j,j}^i = 3e^a$ where $a \sim \mathcal{N}(0, 1), \quad \Sigma_{j,k}^i = 0$ otherwise

| | |
|---|---|
| **Prior** | $\boldsymbol{\theta} = (\theta_1, \theta_2, \theta_3, \theta_4, \theta_5), \quad \theta_i \sim U(-3, 3)$ for $i = 1, \ldots, 5$ |
| **Dimensionality** | $\boldsymbol{\theta} \in \mathbb{R}^5, \boldsymbol{x} \in \mathbb{R}^{100}$ |
| **References** | Greenberg et al. (2019) |

### B.1.3 MA(2) model

The moving average model of order 2 (MA(2)) is a univariate time series model often used as a toy example in the ABC literature. We assign the true parameter values as $\theta = (0.6, 0.2)$ and simulate $x_0 \in \mathbb{R}^{100}$. In this model, $x_o$ and $x$ are high-dimensional, so we use the first two lags of the autocovariance function and the variance (lag 0) as the summary statistics. The preconditioning step required 18k simulations, as the target MCMC acceptance rate is set at 12%. Given that the dimension of the parameter space is relatively low, we employ KDE as an additional unconditional density estimator, utilizing a Gaussian kernel with a bandwidth selected by Silverman method. This method achieves results similar to those obtained using an unconditional normalizing flow.

| | |
|---|---|
| **Simulator** | $x = (x_1, \ldots, x_{100}), \quad x_t = e_t + \theta_1 e_{t-1} + \theta_2 e_{t-2}, \quad$ where $e_t \sim \mathcal{N}(0, 1)$ |
| **Prior** | $\theta = (\theta_1, \theta_2), \quad \theta_i \sim U(-1, 1)$ for $i = 1, 2$ |
| **Dimensionality** | $\theta \in \mathbb{R}^2, \quad x \in \mathbb{R}^{100}$ |
| **Summary Statistics** | Sample autocovariances $S_j(x) = \dfrac{1}{T} \sum_{t=1+j}^{T} x_t x_{t-j},$ for $j = 0, 1, 2$ |
| **References** | Marin et al. (2012) |

### B.1.4 Univariate g-and-k model

The univariate g-and-k model is a popular benchmark in the statistical SBI literature. The four parameters $\theta = (A, B, g, k)$ control the location, scale, skewness, and kurtosis, respectively. We assign the true parameter values as $\theta = (3, 1, 2, 0.5)$ and use them to simulate $x_o \in \mathbb{R}^{50}$ independent observations. For the preconditioning step, we take the set of octiles as summary statistics as they are a robust measure of skewness and kurtosis (Drovandi & Pettitt, 2011). Setting the target MCMC acceptance rate to 20% results in 18k model simulations for the preconditioning step. Here we used a larger acceptance rate to ensure that the preconditioning step does not use too many model simulations.

| | |
|---|---|
| **Simulator** | $x = (x_1, \ldots, x_{50}),$ where $x_i = a + b(1 + c \cdot \tanh[z/2](1 + z^2)^k z, z \sim N(0, 1)$ and $c = 0.8$ |
| **Prior** | $\theta = (a, b, g, k)$ and each parameter has an $U(0, 10)$ prior |
| **Dimensionality** | $\theta \in \mathbb{R}^4, x \in \mathbb{R}^{50}$ |
| **References** | Allingham et al. (2009); Drovandi & Pettitt (2011b) |

### B.1.5 Toad model

The toad movement model proposed in (Marchand et al., 2017) is an individual-based model for simulating the dispersal of Fowler's toads (*Anaxyrus fowleri*). It has been considered as a test example several times in the SBI literature (e.g. Frazier et al. (2022a;b); Drovandi & Frazier (2022)).

This model captures two known behaviours of amphibians: the tendency to return to previously visited locations and a small chance of long-distance movement. Dispersal distance is modeled with a Lévy alpha-stable distribution characterized by a stability factor $\alpha$ and scale factor $\gamma$. The Lévy alpha-stable is symmetrically centred around zero but has heavy tails, allowing the simulation of both frequent short-distance and rare long-distance movements. Toads are modeled to remain at their refuge site during the day and move to forage independently at night. After foraging, the toad may either remain at their current location or return to a previously visited refuge site. We follow "model 1" in Marchand et al. (2017), where each former refuge site has equal probability of being returned to. The probability of a toad returning to a former refuge site is a constant probability, denoted as $p_0$. Hence, there are three parameters to infer, that is $\theta = (\alpha, \gamma, p_0)$.

The simulated data generates an observation matrix (representing Euclidean distance in metres from the origin) of 66 toads across 63 days. We assign the true parameter values as $\theta = (1.7, 35.0, 0.6)$ and use them to simulate $x_o \in \mathbb{R}^{63 \times 66}$. Since the data can be considered as high-dimensional, to summarise the data we first construct four sets of displacement vectors with time lags of 1, 2, 4 and 8 days. For each set, we take the log difference between the $0, 0.1, \ldots, 1$ quantiles, the number of absolute displacements less than 10m, and the median of the absolute displacements greater than 10m, resulting in a total of 48 summary statistics (12 for each time lag).

| | |
|---|---|
| **Prior** | $\alpha \sim U(0, 1), \quad \gamma \sim U(0, 100), \quad p_0 \sim U(0, 0.9)$ |
| **Dimensionality** | $\theta \in \mathbb{R}^3, \quad x \in \mathbb{R}^{63 \times 66}, \quad S(x) \in \mathbb{R}^{48}$ |
| **References** | Marchand et al. (2017) |

## B.2 High-dimensional SVAR model

We consider our illustrative example in a high-dimensional setting with $k = 20$, which leads to 21 parameters that need to be estimated. We set the true parameter values as follows:

$$\theta = (-0.2764, -0.7765, 0.8231, -0.1972, -0.2254, 0.6334, 0.4495,$$
$$0.4465, -0.8961, 0.0647, -0.1791, 0.0795, -0.5464, -0.9354,$$
$$-0.4639, -0.7851, -0.6833, -0.1408, 0.7032, 0.8321, 0.1000),$$

and use these values to simulate the observation dataset. We employ the same summary statistics as in the low-dimensional example and use a uniform distribution as the prior, constrained between -1 and 1 for the $k$ parameters and between 0 and 1 for $\sigma$.

Since the BSL method is the gold standard for this example, we use the standard BSL method proposed by Price et al. (2018), with the total number of simulations set to 20 million.

## B.3 BVCBM

The BVCBM simulation begins by initializing a square domain with cells arranged in a hexagonal lattice. The cell at the center of the domain is identified as a cancer cell, while the others are designated as healthy cells. The simulation proceeds until the tumor reaches a volume of $100\,mm^2$, in accordance with experimental measurements (Wade, 2019; Kim et al., 2011). When this volume is attained, the distribution of healthy and cancerous cells within the lattice is recorded. This configuration then serves as the starting point to

simulate tumor growth over the desired number of days. The model progresses by determining whether a cancer cell proliferates at each timestep, as described by the equation:

$$p_d = p_0 \left(1 - \frac{d}{d_{\max}}\right), \tag{9}$$

where $p_d$ is the probability of cell division, $p_0$ is the initial division rate, $d$ is the current cell density, and $d_{\max}$ is the maximum density. For cancer cells that do not proliferate, the model assesses their potential to transition into invasive cells, governed by the probability $p_{\mathrm{psc}}$. Subsequently, the positions of all cells, both healthy and cancerous, are updated using Hooke's law:

$$\boldsymbol{r}_i(t + \Delta t) = \boldsymbol{r}_i(t) + \frac{1}{\mu}\mathbf{F}_i(t)\Delta t = \boldsymbol{r}_i(t) + \lambda \sum_{\forall j} \frac{\boldsymbol{r}_{i,j}(t)}{\|\boldsymbol{r}_{i,j}(t)\|}(s_{i,j}(t) - \|\boldsymbol{r}_{i,j}(t)\|). \tag{10}$$

Here, $\boldsymbol{r}_i(t + \Delta t)$ denotes the updated position of cell $i$, $\mu$ is the cell motility coefficient, $\mathbf{F}_i(t)$ is the force on cell $i$, $\lambda$ is a mechanical interaction coefficient, $\boldsymbol{r}_{i,j}(t)$ is the vector between cells $i$ and $j$, and $s_{i,j}(t)$ is the natural length of the spring connecting the two cells. The parameters for the mechanical interactions, such as $\lambda$ and $\mu$, are sourced from prior studies in the literature Meineke et al. (2001). See Jenner et al. (2020); Wang et al. (2024) for more detailed model simulation.

Four parameters $\theta = (p_0, p_{\mathrm{psc}}, d_{\max}, g_{\mathrm{age}})$ control the tumor growth during a single phase, which is a period when the tumor grows based on fixed values for these four parameters. For the biphasic model, an additional parameter $\tau$ is introduced, representing the time at which the tumor growth pattern changes, that is, the values for $\theta$ change. Therefore, for BVCBM, we need to estimate nine parameters for two pancreatic cancer datasets, denoted as $\theta_1 = (p_0^1, p_{\mathrm{psc}}^1, d_{\max}^1, g_{\mathrm{age}}^1)$, $\theta_2 = (p_0^2, p_{\mathrm{psc}}^2, d_{\max}^2, g_{\mathrm{age}}^2)$, and $\tau$, so that $\theta = (\theta_1, \theta_2, \tau)$.

The parameter $p_{\mathrm{psc}}$, which is the probability of tumor cell invasion into healthy cells, significantly affects the simulation time. The value of $p_{\mathrm{psc}}$ should be around $10^{-5}$, indicating that an increase in probability will require more cells to be simulated. Moreover, a smaller value of $p_{\mathrm{psc}}$ results in simulation time. For PNPE, the total simulation time for 15k simulations (26-day dataset) and 17k simulations (32-day dataset) for the preconditioning step is approximately 13 and 15 minutes, respectively, whereas SNPE requires around 1 hour for the first round (i.e. based on samples from the prior) of 10k simulations. This is because the preconditioning step is effective at quickly eliminating values of $p_{\mathrm{psc}}$ that lead to longer model simulation times.

## C   Further experimental results

### C.1   SVAR

We generate 10 different datasets for the SVAR model in 6 dimensions based on predefined parameter values shown in Table 3. Then, we run SNPE and PNPE with 10 different random seed values for each dataset to investigate reproducibility. For preconditioning, we use a 10% stopping rule and train unconditional normalizing flows based on ABC posterior samples for different random seed values for each dataset. To quantify the results, we use MMD as the metric. For each dataset, we compute the MMD values between the approximate distribution and the reference distribution, allowing us to use one value to summarize the fit. By computing the MMD for the approximate posterior based on each set of random seed values, we produce a boxplot corresponding to each method.

| DATASET | $\theta_1$ | $\theta_2$ | $\theta_3$ | $\theta_4$ | $\theta_5$ | $\theta_6$ |
|---|---|---|---|---|---|---|
| 1 | 0.9749 | -0.2564 | 0.4230 | 0.8749 | 0.6453 | -0.1772 |
| 2 | 0.2777 | -0.2932 | 0.5483 | -0.4679 | 0.1549 | -0.6365 |
| 3 | 0.3193 | -0.2509 | 0.7801 | 0.5103 | -0.4358 | 0.0317 |
| 4 | -0.9295 | -0.3267 | 0.2930 | -0.9326 | -0.6113 | -0.5015 |
| 5 | 0.5146 | 0.2805 | -0.6695 | 0.7410 | -0.4518 | -0.4366 |
| 6 | 0.6401 | -0.6037 | 0.3730 | -0.6739 | -0.7237 | -0.5494 |
| 7 | -0.9378 | -0.6438 | -0.8434 | -0.9219 | -0.4032 | 0.5577 |
| 8 | -0.2663 | 0.9098 | 0.7383 | 0.1438 | -0.0968 | 0.7992 |
| 9 | 0.9185 | -0.1454 | 0.5147 | -0.2921 | 0.0595 | 0.7227 |
| 10 | -0.4766 | 0.7152 | 0.9845 | -0.7105 | -0.2426 | -0.4923 |

Table 3: **True values to generate observational datasets:** Each row refers to the true parameters that we used to generate the observation datasets $x_o$ with the additional parameter $\sigma$ set to a constant value of 0.1.

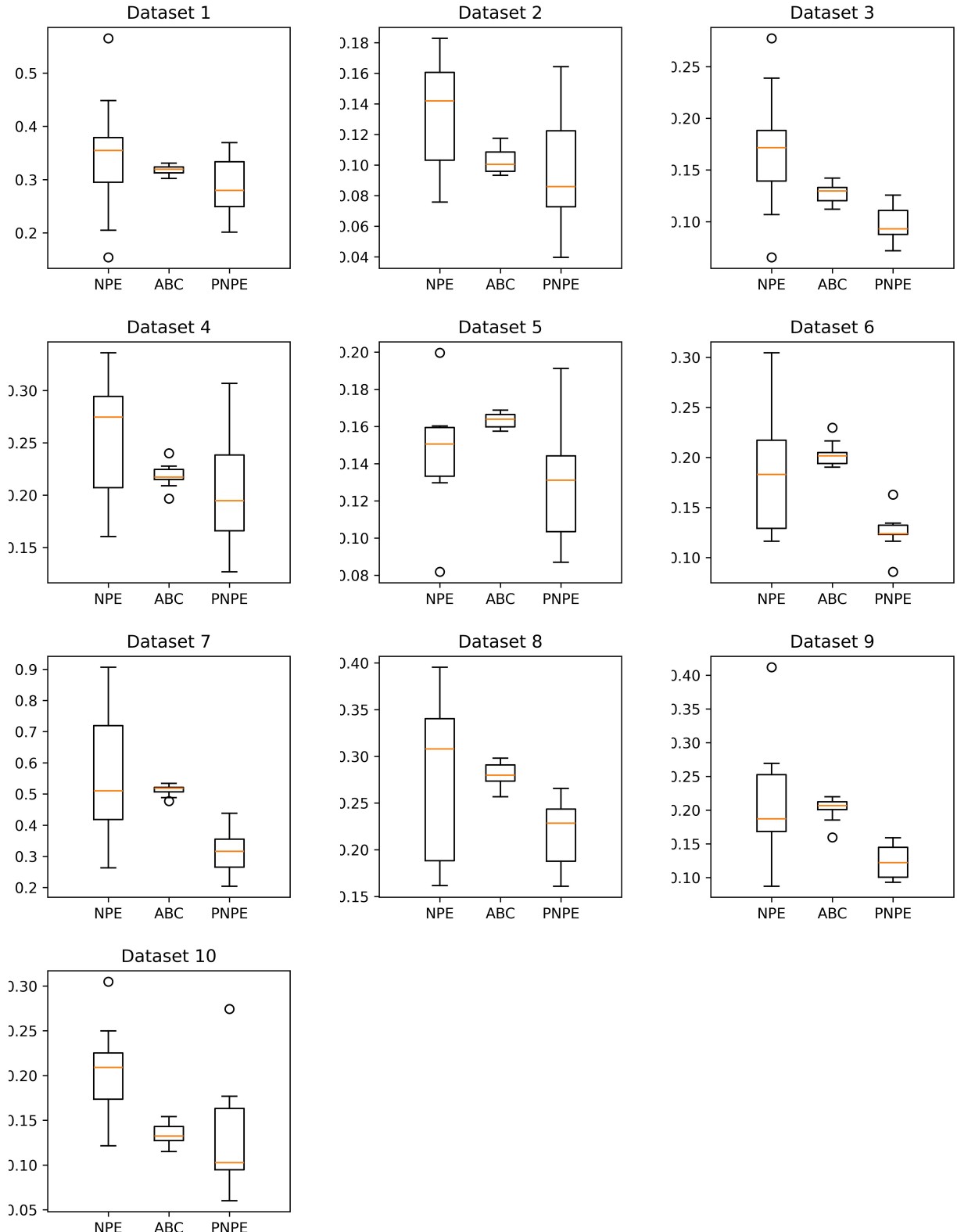

Figure 10: **Performance for SVAR with 10 different datasets:** The box-plots refer to the MMD values computed from approximate posterior for each algorithm with 10 different random seed values.

## C.2   BVCBM

In this section, we present prior predictive distributions of tumour volumes in (a) and (b) for two pancreatic datasets (in (c) and (d) we show the same plots but on the log scale). We also show the posterior predictive distributions on the log scale obtained with different methods in Figure 12. Then we present the bivariate posterior density and marginal posterior density plots for the BVCBM as additional results. It is evident from Figures 13 and 14 that PNPE provides more precise estimation than SNPE for both pancreatic cancer datasets.

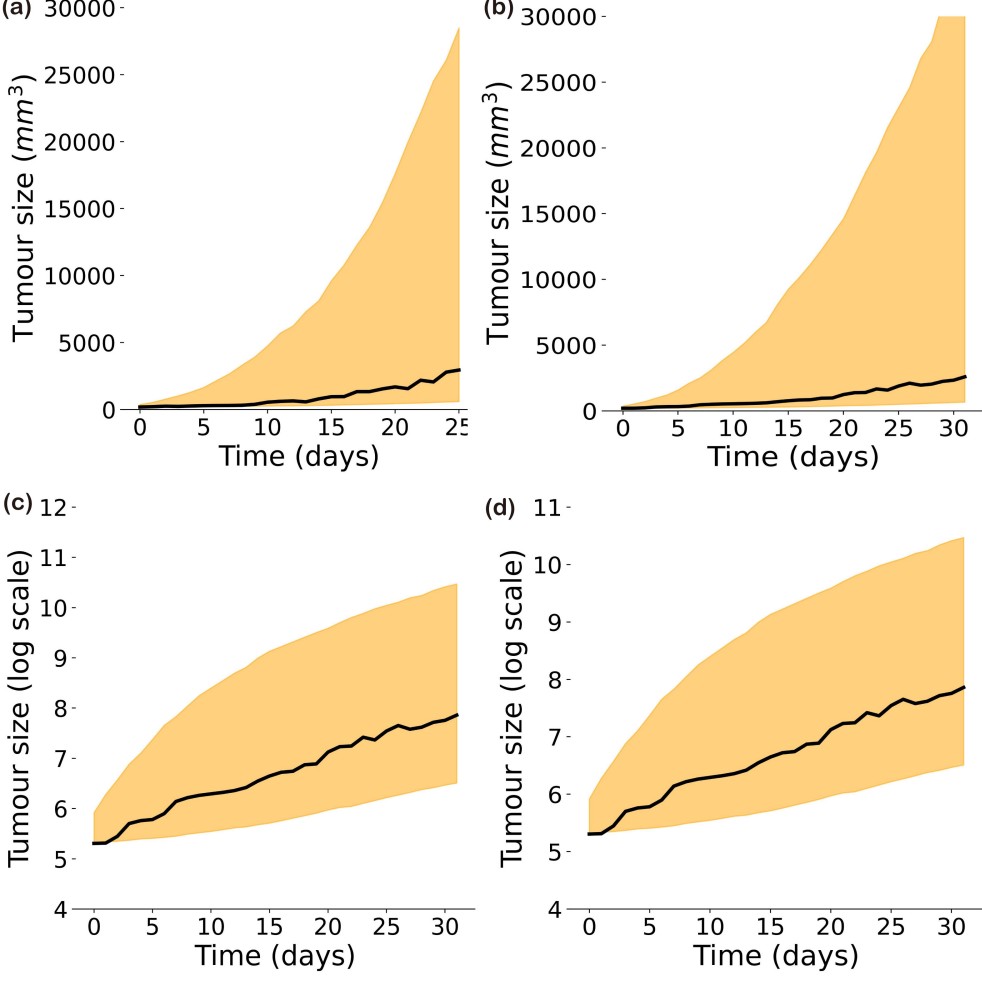

Figure 11: **Prior predictive distribution for BVCBM.** We sample 10k parameter values from prior distribution and plot the prior predictive distribution for two pancreatic datasets. In (a) and (b), the plots are in regular scale, and in (c) and (d), the plots are in log scale.

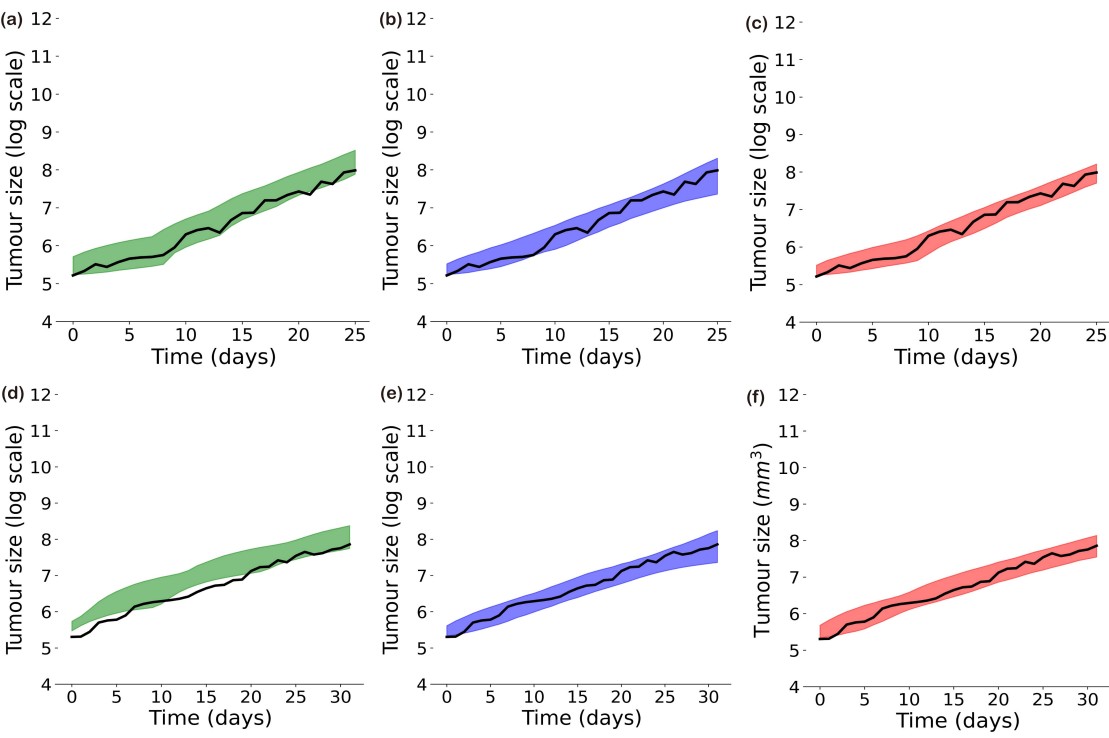

Figure 12: **Posterior predictive distributions for two pancreatic cancer datasets in log scale.** 90% Posterior predictive interval plots for SNPE (**a**), preconditioning step (**b**) and PNPE (**c**) for the 26-day dataset; 90% Posterior predictive interval plots for SNPE (**d**), preconditioning step (**e**) and PNPE (**f**) for the 32-day dataset.

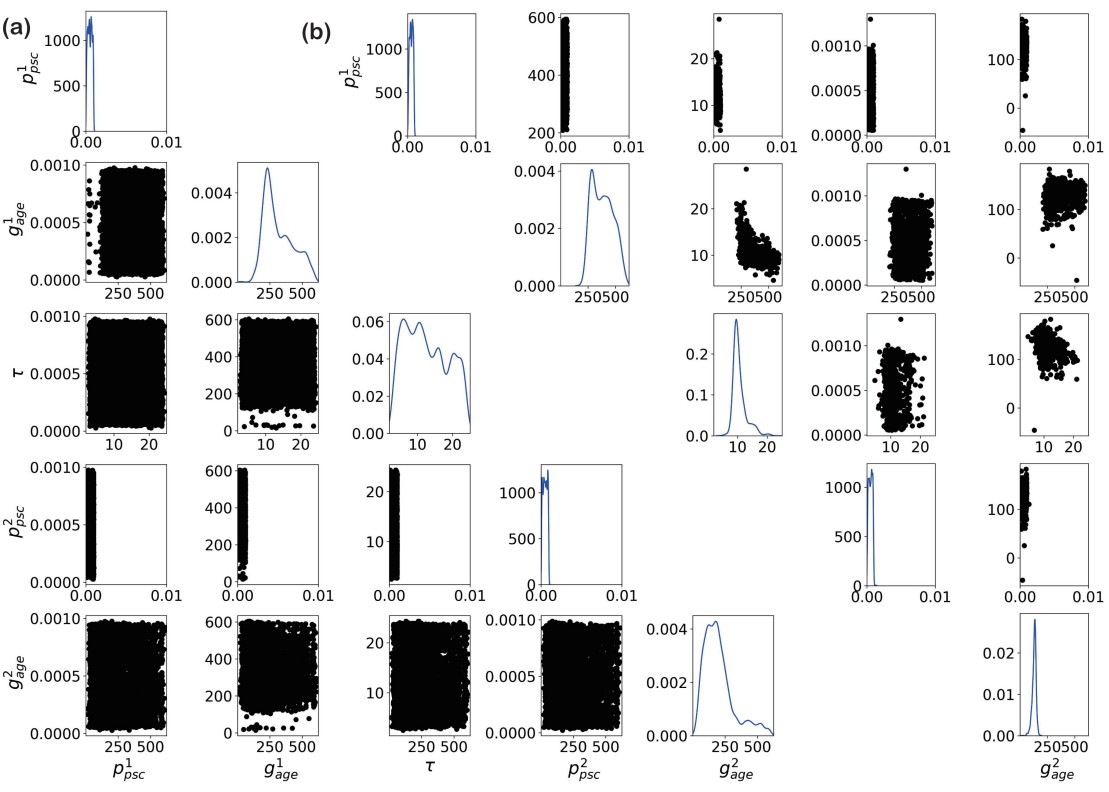

Figure 13: Bivariate density plots for the pancreatic dataset with 26-day measurements for (a) SNPE and (b) PNPE. The diagonal entries represent the marginal posterior densities for $(p^1_{\text{psc}}, g^1_{\text{age}}, \tau, p^2_{\text{psc}}, g^2_{\text{age}})$.

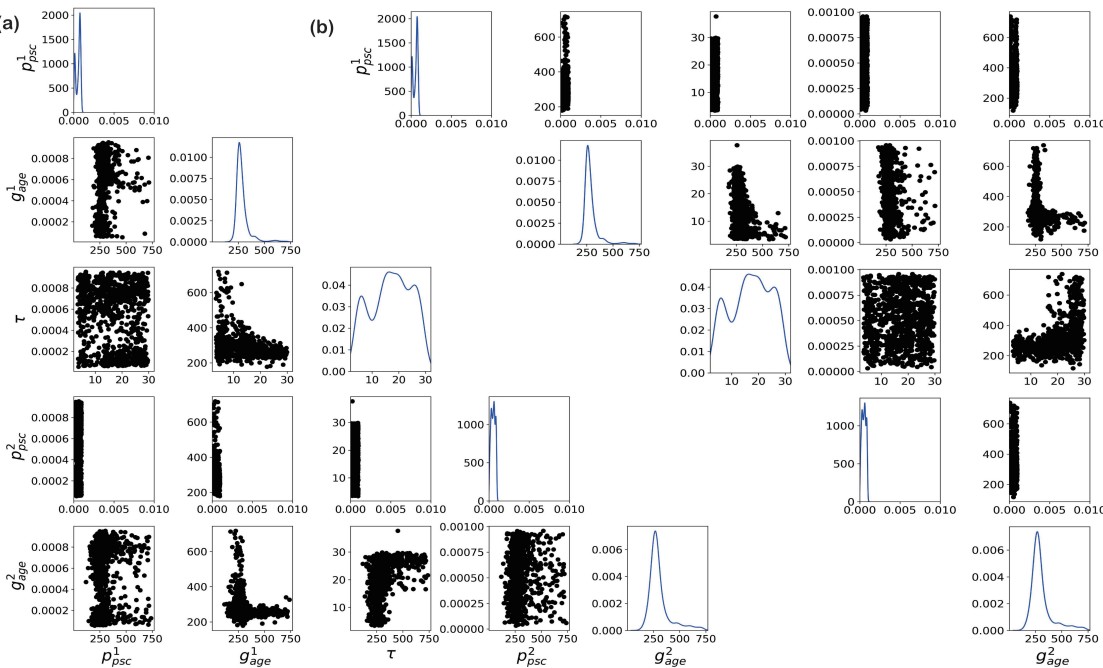

Figure 14: Bivariate density plots for the pancreatic dataset with 32-day measurements for (a) SNPE and (b) PNPE. The diagonal entries represent the marginal posterior densities for $(p_{psc}^1, g_{age}^1, \tau, p_{psc}^2, g_{age}^2)$.

