# OpenReview forum: "Preconditioned Neural Posterior Estimation for Likelihood-free Inference"
_TMLR — Accepted by TMLR_

### Review · Reviewer_s2up · 2024-07-02

**Summary Of Contributions:**

In this work, the authors propose a preconditioning method for neural posterior estimators (NPE) that works by (1) running several iterations of approximate Bayesian computation (ABC) to get a rough estimate of the parameters used to generate data, (2) fitting an unconditional density estimator on the ABC output, and (3) using this as the prior in the computation of the NPE. They demonstrate two examples where standard NPE might fail and show that by preconditioning, this can be fixed.

**Audience:**

Yes

**Broader Impact Concerns:**

Not applicable.

**Claims And Evidence:**

Yes

**Requested Changes:**

- The sentence "A neural density estimator ... used for NPE that trained by minimise the loss" in page 3 is incoherent and should be fixed.
- The acronym "BSL" in page 4 and thereafter is not introduced.
- The acronym "TSNPDE" in Section 4 is only introduced briefly in an earlier section, which is easy to miss. It might be worth reintroducing what the "T" in TSNPDE stands for again in Section 4 just to be clear.
- In some places, the authors mention difficulties due to "leakage issues". What does this mean exactly?

**Strengths And Weaknesses:**

__Strengths:__

Overall, the paper is well-written and easy to follow. The algorithms are explained clearly and with sufficient details (especially in the appendix). The two experiments show promising effect that the proposed preconditioning method has on likelihood-free parameter inference.

__Weakness:__

- The method that is proposed, by combining ABC and NPE, is really quite simple and difficult to say whether it is completely novel. However, with TMLR's emphasis on technical correctness over novelty, this is not a major issue.
- In the experiments, it would be better to have further explorations of the cost-accuracy trade-off. For example by showing e.g. exactly how many iterations of ABC should be run before observing improvements over vanilla (S)NPE and how much cost that would incur. Also, the experiments only show qualitative results on a select few examples, which puts into question the robustness of the method. By using a metric to quantify the resulting posterior estimate when compared to the output of BSL (e.g. using the Wasserstein distance, maximum mean discrepancy, KL divergence, etc...), and displaying the scores with error bars (e.g. arising from different random seeds for MCMC sampling or neural network initialisation), we would have a better understanding of how robust the method is. Also you can quantify the results in Figure 7 rather easily using e.g. negative log-predictive density, and present the scores across multiple random seeds.
- I am also not sure if "preconditioning" is the right phrase to describe the proposed method. To my knowledge, "preconditioning" refers specifically to methods for improving the condition number of a matrix in a linear system and I'm not sure how generally this term could be applied.

---

> ### Author Response · Authors · 2024-07-30
> **Thanks comments from reviewer s2up**
>
> We thank the reviewer for taking the time to write this review. We appreciate the reviewer's suggestions to further explore the cost-accuracy trade-off of the preconditioning step. We have added a new section in the Methods section and compared the performance of PNPE by running it with different choices for the percentage of the stopping rule. We find that if the percentage is too large, ABC will stop too early, leading to a loss of the benefit of the preconditioned part. If the percentage is too small, it will result in ABC running too long and using too many model simulations. To investigate reproducibility, we run multiple runs of each algorithm with 10 different datasets each with 10 different random seeds for the SVAR model in dimension 6. We present those results in Appendix C1. For each dataset, we compute 10 maximum mean discrepancy (MMD) values between the reference distribution and approximate distribution based on different random seeds so that we can use one value to summarize the fit. Then, we produce a boxplot for each algorithm for each dataset.  We find generally across datasets that PNPE is the best performing method and never produces poor results.
>
> We have provided a detailed explanation of the leakage problem and indicate that truncated SNPE (TSNPE) is an approach to use truncated proposal distribution to avoid the leakage problem. We noticed that for the SVAR model, the likelihood function is available so it is easy to obtain exact posterior samples, so we do not use Bayesian synthetic likelihood (BSL) results anymore.
>
> We thank the reviewer for questioning the suitability of word ‘preconditioning’. Preconditioning is also used in inference context, e.g., [1-2], to indicate the preprocessing of an inference method. Our approach is to use the ABC algorithm to localize the high-density regions and then run NPE. So we can interpret this as a preconditioning step for running NPE.
>
> Finally, thank you for noticing the minor issues; we have now addressed all of these. We use blue text to indicate the changes or new content in the manuscript.
>
> ### Reference:
> [1]: Agapiou, S., & Mathé, P. (2014). Preconditioning the prior to overcome saturation in Bayesian inverse problems. arXiv preprint arXiv:1409.6496.
>
> [2] Siahkoohi, A., Rizzuti, G., Louboutin, M., Witte, P. A., & Herrmann, F. J. (2021). Preconditioned training of normalizing flows for variational inference in inverse problems. arXiv preprint arXiv:2101.03709.

---

### Review · Reviewer_MRjJ · 2024-07-07

**Summary Of Contributions:**

The paper proposes a method for Likelihood free inference (LFI) combining ABC and SNPE. Typically, the first round of SNPE involves running NPE, and the resulting approximation is used as proposal distribution for SNPE’s second round. The approach proposed here builds a different first approximation, consisting of fitting a normalizing flow to the samples produced by some ABC method. Overall, I find the technical contribution of the paper limited, though it proposes a clever way of using ABC in concert with SNPE.

**Audience:**

Yes

**Claims And Evidence:**

Yes

**Requested Changes:**

Overall, I find the technical contribution of the paper limited, though it proposes a clever way of using ABC in concert with SNPE. A few comments / questions.

- Typo in section 2.2. Very similar paragraphs repeated.

“…utilizing a neural network F and its adjustable network weights ψ, is often used
as an NPE. In order to train qF (x,ψ)(θ), the following loss is minimized:

A neural density estimator qF (x,φ)(θ) used neural network F with adjustable network weights φ is used for NPE that trained by minimize the loss:..."

- It would be good to explain the SVAR example more thoroughly. For instance, X is a 6 by 6 matrix, what are the elements of X that are non-zero and exact mapping to theta? It is also not totally clear to me what’s the size of the generated dataset to train NPE. It is also mentioned that the lag-1 autocorrelation is used as summary statistic. It is also not clear to me what the size of this summary statistic is. Since we have the lag 1 autocorrelation for each pair (i, j), are the observations 36-dimensional?

- NPE-based methods do not really need the summary statistic, as they can “learn their own” with the trained network, which may be beneficial. The summary statistic is needed by ABC methods. So I’d think a comparison against NPE that takes in the raw data would be a good addition.

- Overall I find that the empirical evaluation could be extended. Including a few more models would be good. Specially, since NPE has been observed to perform well in a range of settings (for instance, a few examples from the “benchmarking SBI methods” paper could be used). Additionally, I’d be interested in seeing comparisons against plain NPE in all models, where NPE is trained using the same number of model simulations as the total amount used by other methods (adding across rounds).

**Strengths And Weaknesses:**

**Summary:** The paper proposes a method for Likelihood free inference (LFI) combining ABC and SNPE. Typically, the first round of SNPE involves running NPE, and the resulting approximation is used as proposal distribution for SNPE’s second round. The approach proposed here builds a different first approximation, consisting of fitting a normalizing flow to the samples produced by some ABC method.

The method’s practical usefulness comes down to whether the first approximation provided by ABC is better than the first approximation provided by NPE (in the context of SNPE).

Why may this be the case? In principle, ABC methods are known to suffer from extremely slow convergence, often slower than NPE. However, in this case, since the ABC approximation will be refined by a subsequent application of SNPE, the ABC method does not need to operate on the challenging regime of epsilon->0 (which has a big effect on the method’s efficiency).

All in all, if my understanding is correct, the claim in the paper, that the proposed method is useful in practice, can be re-interpreted as follows: The first approximation obtained by fitting a normalizing flow to ABC produced samples (in a “relaxed” configuration with epsilon>>0) is better than the first approximation provided by NPE (i.e. fitting an amortized normalizing flow with samples from the prior and the simulator). This needs to be verified empirically over a range of models and settings (see more below on requested changes, specifically empirical evaluation).

**Method benefits:** The approach seems sensible when we have a single observed dataset (i.e. we are not interested in amortization, as the method cannot handle that scenario). Running NPE as a first step of SNPE may be tackling a harder task than necessary, as NPE yields an amortized approximation that can be used for any potential observation. ABC, on the other hand, focuses on producing an approximation for the set of observations at hand, a potentially simpler problem to solve. Additionally, ABC used as proposed in this work does not need to yield a perfect approximation (which is hard), as its results will be refined with SNPE. It should be useful as long as it provides a better approximation than plain NPE. As mentioned above, ABC does not need to operate in the challenging epsilon->0 regime, as its output will be refined by further rounds of SNPE.

**Method drawbacks:** Reliance on convergence of ABC (for a fixed level of epsilon). While convergence may be easier to achieve for epsilon>>0, if ABC leaves important regions of the parameter space unexplored, the method will likely fail to yield good results (same holds for normal SNPE. If the first approximation provided by NPE is not a good one).

---

> ### Author Response · Authors · 2024-07-30
> **Thanks comments from reviewer MRjJ**
>
> We thank the reviewer for taking the time to write this review. We appreciate the reviewer pointing out the necessity of extending the empirical evaluation. We have added five more models, two from the benchmarking tasks in Lueckmann et al. (2021) and three others from the ABC literature. For the two-moon and SCLP models, we find that there are no extreme values in the simulated datasets, leading SNPE to perform better. For the other three models, g-and-k and the toad model, we find that the prior predictive distributions are complex and NPE might not perform well, especially for the g-and-k model. For g-and-k model, we use full datasets instead summary statistics. We have added a new section to present the results for benchmarking tasks in the Further Experiments section. Furthermore, in Appendix C1, we run each algorithm with different random seeds for 10 observational datasets for the 6-dimensional SVAR example, where such datasets are generated based on predefined parameter values in Table 3.
>
> We thank the reviewer for pointing out that NPE does not require summary statistics, as (S)NPE can learn their own. This is a good point that NPE-based approaches can learn their own “summaries”/low-dimensional mapping which is a very useful aspect of these methods.  However, such a summary network would still need to take extreme simulated data as input, not obvious to us that this would learn a reasonable low-dimensional mapping - at least in the considered experiments where the summaries were carefully chosen. We also note, some ABC methods do not require summarisation, e.g., Wasserstein ABC and K2-ABC. Our preconditioned method would work the same with such summary-free ABC methods with NPE with a summary network, but we leave this for future work and make a note of it in the discussion section.
>
> We have added a more detailed explanation for the SVAR model in section 2.2.2 and addressed all minor issues by using blue text to indicate changes or new content.

---

> > ### Comment · Reviewer_MRjJ · 2024-08-05
> > **Thank for you response**
> >
> > Thanks you for answering my comments and adding new empirical results as suggested in the review. I think these improve the paper.

---

### Review · Reviewer_bPPq · 2024-07-18

**Summary Of Contributions:**

The authors describe an approach to set good initial values to Neural Posterior Estimation approaches.

**Audience:**

No

**Claims And Evidence:**

No

**Requested Changes:**

Requested Changes

“... we run three rounds of SNPE and compare the results with those from BSL, considering BSL results as the gold standard for this example…” What is BSL? Can you please describe this method and associated works?

“...We use summary statistics to reduce the dimension of the data. Following Thomas et al. (2020); Drovandi et al. (2023), we use the lag 1 autocovariance 1 T PT t=2 y i t y j t−1 as the summary statistics,...” What is the role of the summary statistics in NPE? I only see the importance defined for ABC.

I generally find Figures 1-6 to be weak and lacking critical interpretation. Ideally, all of these figures would be in the supplement, and a more parsimonious summary and interpretation is provided by the authors to the reader.

Are there more complex examples that create multimodal data distributions in one mode is very rare. In the ABC sampling approach, would these parameter values be found initially? If not, would this approach capture it?

“However, APT can suffer from a ‘leakage’ issue….” and “We observed a leakage problem occurring in rounds 8 and 5 for the 26-day and 32-day datasets, respectively” What do you mean by a leakage problem? Mathematically, why does this solve the leakage problem? This seems like a pretty important choice for the cancer dataset analysis.

“Figure 7: Posterior predictive distributions for two pancreatic cancer datasets. 90% Posterior predictive interval plots for SNPE (a), preconditioning step (b) and PNPE (c) for the 26-day dataset; 90% Posterior predictive interval plots for SNPE (d), preconditioning step (e) and PNPE (f) for the 32-day dataset.” Where is the comparison with a standard reference approach? What about BSL (which hasn’t been described).

Is there any biological interpretation to your results? Is the modeling good or bad? What can we do with these results?

**Strengths And Weaknesses:**

Strengths

I think this is a pragmatic approach to setting the prior when using NPE.

The authors use a model system that has been characterized previously and is relatively tractable.

“Finally, we consider a challenging real-world problem in cancer biology: calibrating the biphasic Voronoi cell-based model (BVCBM) (Wang et al., 2024) that models tumor growth.” I think this is a very interesting problem!

Weaknesses

“ NPE uses N training pairs of simulator parameter values and simulated datasets, {θi , xi} N i=1, to estimate the posterior distribution p(θ|x) (Papamakarios et al., 2021)” I’m confused why Papamakarios 2021 is the reference for this work, as it does not explicitly define Neural Posterior Estimation, nor do I readily see this formula. Generally, where does this simulated data and parameters come from?

BSL needs to be defined. Since it is used as the reference for this work, it’s impossible to compare advances described herein.

I don’t know how to interpret what is “good” from FIgures 1-6. Moreover, the results are entirely subjective. Is there any summary of the different approaches? What is “good” for posterior predictive distributions and marginal posterior distributions? In particular, Figures 5 and 6 are not interpretable.

In the ABC preconditioning step, what is the distribution of the initial parameters? How does it differ than the priors used previously?

“Finally, we consider a challenging real-world problem in cancer biology: calibrating the biphasic Voronoi cell-based model (BVCBM) (Wang et al., 2024) that models tumor growth.” What is the literature in optimizing BVCBM? Is it standard Bayesian inference? Why is your method a good and reasonable approach here?

---

> ### Author Response · Authors · 2024-07-30
> **Thanks comments from reviewer bPPq**
>
> We thank the reviewer for taking the time to write this review. We appreciate the reviewer pointing out the limitations of the figures, and we would like to address this point here. For the posterior distribution plots, such as Figure 3, PNPE can produce more precise posteriors concentrated around the true values compared to other methods. By showing posterior predictive plots, such as Figure 4, we can demonstrate that it is not over-concentrated by using repeated simulation results. We define "good" for posterior predictive simulation plots as the predictive intervals being as narrow as possible while still covering the majority of the data (e.g., roughly 95%). For example, in Figure 9, we can see that PNPE performs better since the posterior predictive distribution is the narrowest but still captures most or all of the data.
>
> We thank the reviewer for pointing out the impact of summary statistics in NPE. For the SVAR model, summary statistics are used as default for the demonstration purpose. For the newly added benchmarking example, g-and-k, which is a popular example in the SBI literature, we use the full datasets instead of summary statistics. It is possible to use some distance functions, such as Wasserstein ABC [1] and K2-ABC [2], in the ABC preconditioning step to use the full dataset if desired.
>
> We have provided a more detailed explanation of the leakage problem and indicate that truncated SNPE (TSNPE) is an approach using a truncated proposal distribution to avoid the leakage problem.
>
> Since the exact posterior is available for the SVAR example, we no longer use Bayesian synthetic likelihood (BSL) to obtain the reference distribution. For the BVCBM examples, we do not use BSL to get the reference distribution since we do not know how the real datasets were generated. Instead, we use posterior predictive checks to verify if the predictive distribution fits the real tumor growth datasets well. We refer to [3-4] for details of BVCBM and its application in cancer biology.
>
> We use blue text to indicate the changes or new content in the manuscript.
>
> ### Reference:
> [1]: Bernton, E., Jacob, P. E., Gerber, M., & Robert, C. P. (2019). Approximate Bayesian computation with the Wasserstein distance. Journal of the Royal Statistical Society Series B: Statistical Methodology, 81(2), 235-269.
>
> [2]: Park, M., Jitkrittum, W., & Sejdinovic, D. (2016, May). K2-ABC: Approximate Bayesian computation with kernel embeddings. In Artificial intelligence and statistics (pp. 398-407). PMLR.
>
> [3]: Wang, X., Jenner, A. L., Salomone, R., Warne, D. J., & Drovandi, C. (2024). Calibration of agent based models for monophasic and biphasic tumour growth using approximate Bayesian computation. Journal of Mathematical Biology, 88(3), 28.
>
> [4]: Jenner, A. L., Frascoli, F., Coster, A. C., & Kim, P. S. (2020). Enhancing oncolytic virotherapy: Observations from a Voronoi Cell-Based model. Journal of Theoretical Biology, 485, 110052.
>
> [5]: Jenner, A. L., Kelly, W., Dallaston, M., Araujo, R., Parfitt, I., Steinitz, D., ... & Vine, K. L. (2023). Examining the efficacy of localised gemcitabine therapy for the treatment of pancreatic cancer using a hybrid agent-based model. PLOS Computational Biology, 19(1), e1010104.

---

### Decision · Action_Editor_LsQG · 2024-08-26

**Recommendation:** Accept with minor revision

**Comment:**

The paper unfortunately could use considerably more polish. Please address the following in the revised manuscript:
* Proofread the paper for typos and correct them, there is a fair number of them.
 * Correct "neural posterior estimator" to "neural posterior estimation" in the abstract.
 * The paragraph describing the leakage problem in Section 2.2 is very unclear. Please rewrite it to explain clearly what exactly is being leaked and why, avoiding vague phrases like "untrained areas of neural network".
* The description of the transition matrix in Section 2.2.1 is either incomplete or incorrect, as the claim that there is one non-zero off-diagonal element per column does not follow from conditions (1) and (2). Perhaps there's an unstated assumption that each variable is coupled to only one other variable? Also, condition (1) seems redundant given the sentence preceding it.
 * Make it clear how many rounds were used for (T)SNPE for the BVCBM.
 * Clarify that the claim about q recovering the true posterior after Eq. 2 is true only in the limit of infinite data.
 * Make the description of the SMC ABC algorithm more precise and explain what parameter c is supposed to be.

**Audience:**

The investigation of using ABC-generated samples instead of samples from the prior in the first round of SNPE is interesting enough to have a TMLR audience.

**Claims And Evidence:**

The claims are sufficiently supported.

---

> ### Author Response · Authors · 2024-09-06
> **Many thanks for comments**
>
> Many thanks to the reviewers and action editor for the time spent reviewing our paper and the great suggestions. We have now uploaded a camera-ready revision with all the requested changes.

---

> > ### Comment · Action_Editor_LsQG · 2024-09-17
> > **One more correction request**
> >
> > Dear authors,
> >
> > Thank you for making the requested revisions. While verifying them, I noticed that the captions for many figures (1, 2, 4, 5 and others) state that the ground truth values are shown using black dashed lines, while in reality solid grey lines are used. Please correct this.

---

> > > ### Author Response · Authors · 2024-09-17
> > > **Thanks for comment**
> > >
> > > Many thanks for pointing out the typo in the caption for the SVAR example. We have corrected it to grey solid lines instead of black dashed lines.